# Real-World Reinforcement Learning of Active Perception Behaviors

**Edward S. Hu**[*,1]    **Jie Wang**[*,1]    **Xingfang Yuan**[*,1]    **Fiona Luo**[1]    **Muyao Li**[1]
**Gaspard Lambrechts**[2]    **Oleh Rybkin**[3]    **Dinesh Jayaraman**[1]
[*] Equal contribution    [1]University of Pennsylvania    [2]University of Liège    [3]UC Berkeley

## Abstract

A robot's instantaneous sensory observations do not always reveal task-relevant state information. Under such partial observability, optimal behavior typically involves explicitly acting to gain the missing information. Today's standard robot learning techniques struggle to produce such active perception behaviors. We propose a simple real-world robot learning recipe to efficiently train active perception policies. Our approach, asymmetric advantage weighted regression (AAWR), exploits access to "privileged" extra sensors at training time. The privileged sensors enable training high-quality privileged value functions that aid in estimating the advantage of the target policy. Bootstrapping from a small number of potentially suboptimal demonstrations and an easy-to-obtain coarse policy initialization, AAWR quickly acquires active perception behaviors and boosts task performance. In evaluations on 8 manipulation tasks on 3 robots spanning varying degrees of partial observability, AAWR synthesizes reliable active perception behaviors that outperform all prior approaches. When initialized with a "generalist" robot policy that struggles with active perception tasks, AAWR efficiently generates information-gathering behaviors that allow it to operate under severe partial observability for manipulation tasks. Website: https://penn-pal-lab.github.io/aawr/

## 1 Introduction

Any organism needs to extract information from the world via its sensory apparatus to make decisions, solve tasks, and survive. One strategy is to have high bandwidth and sophisticated sensors like the human eye, to sense as much information as possible and embrace the "blooming, buzzing confusion of the senses" [1] this entails, subject to natural limits, such as a local field of view. Another strategy is to use the ability to move around in the world to gather new information and overcome our sensory limitations - we scan our eyes across a crowded party to find a friend, and polish our glasses to get a clearer view. Such information gathering behaviors are called active [2] or interactive [3] perception based on whether they only move a sensor around the world or if they also alter the world. In the following, we will use "active perception" as shorthand to refer to all such behaviors, except when the distinction is particularly pertinent. In this work, we are interested in learning active perception behaviors to compensate for the limitations of various sensory setups in robots, ranging from entirely blind robots operating purely from proprioception, to robots operating with sophisticated multi-camera setups.

It has been hard to learn useful active perception behaviors for robotics, and not for lack of trying [2–12]. Of the techniques commonly in vogue for robotics, imitation learning is ill-suited because acquiring optimal active perception demonstrations can be cumbersome and unnatural (e.g. forcing a teleoperator to look through wrist cameras). In theory, RL should be able to learn active perception behaviors from interaction, but in practice it is too sample-inefficient even in fully observed settings, leave alone the partially observed settings where active perception is relevant. Moreover, sim-to-real

39th Conference on Neural Information Processing Systems (NeurIPS 2025).

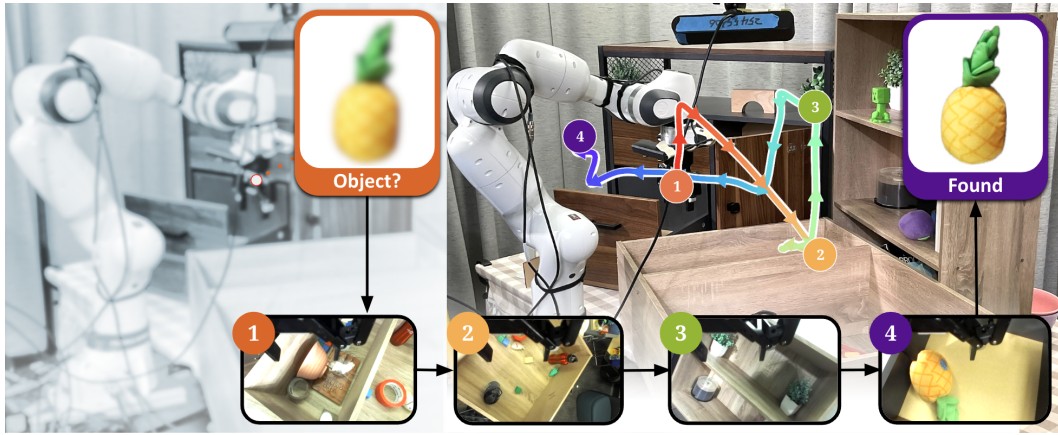

Figure 1: Left: A robot with a single wrist image struggles to find objects in a heavily occluded scene. Right: An active perception policy searches through potential hiding spots to find the target object.

transfer is hard for such tasks because it is tied closely to sensory capabilities, and many sensory observations of interest, including depth, LIDAR, RGB cameras, audio, touch, contact, force sensors, etc., are all hard to simulate well enough to reliably learn transferable behaviors. This difficulty is also reflected in the inability of today's state-of-the-art generalist policies trained on massive amounts of robot tele-operation data to perform even simple search tasks, as we will show in experiments.

We demonstrate that bootstrapping from suboptimal demonstrations and incorporating rewards in an offline-to-online RL algorithm is a viable approach for synthesizing active perception behaviors. Critically, the RL algorithm must be designed appropriately, and we theoretically derive that asymmetric access to extra sensors for critics and value functions is important to correctly estimate supervision signals for the policy in partially observed settings. Specifically, we take a weighted behavior cloning approach, by extending advantage weighted regression (AWR) [13–16] to incorporate privileged, training-time only observations. We call this approach **Asymmetric AWR** (AAWR). As an example, in one experiment, we give critics privileged access to object detectors to train open-loop policies that only receive proprioception and initial object positions. In another experiment, we give critics privileged access to privileged segmentation masks to train search policies in visually cluttered scenes.

Our key contributions are as follows: **(1)** We efficiently train real-world active perception policies by introducing AAWR, which uses privileged value functions to better supervise the policy. **(2)** We provide theoretical justification for using privileged advantage estimates for AWR in POMDPs, by showing that maximizing the expected policy improvement in a POMDP results in the AAWR objective. **(3)** We demonstrate that AAWR effectively learns a variety of active and interactive perception behaviors in 8 different settings - over varying types of partial observability, multiple types of simulated and real robots, and varied tasks.

## 2   Related Work

Active perception policies are frequently trained to optimize information-theoretic objectives such as uncertainty reduction and next best viewpoint selection[17–21]. Such approaches are used for task-agnostic applications like object tracking [8, 22], scene reconstruction [11, 23, 24], pose estimation [12], or free-space navigation [20]. However, such approaches are not applicable to our setting in a variety of ways. First, many assume the ability to freely query views of a scene, without regard to task constraints [17, 19]. In our manipulation settings with clutter, there are many informative viewpoints that are difficult to reach due to physical constraints. Next, such information-theoretic metrics are not task-relevant - to locate a toy, a human may naturally look in drawers, shelves, cabinets, or other storage areas. But these information-theoretic metrics do not incorporate such task information, and may find the unseen back of a shelf just as interesting. In short, for training active perception policies for robots, we desire a more task-centric active perception approach that considers the constraints of the task and does perception-improving behavior to maximize the task success rate, instead of a task-agnostic metric.

Imitation and reinforcement learning are natural ways to synthesize task-relevant active perception policies, by using demonstrations and reward functions. Some prior works[5, 6, 25] use imitation learning to train active perception policies on real robots, but performance is bounded by the demonstrator. However, acquiring optimal active perception demonstrations can be quite cumbersome (e.g. forcing human tele-operators to look through wrist cameras). On the other hand, RL approaches do not impose this burden, but real-world RL approaches for active perception [26, 27] require heavy instrumentation (e.g. for constructing task-specific volumetric maps), limiting generality. Without such assumptions, RL methods are often limited to simulation due to sample inefficiencies [18, 28, 29]. Sim2real transfer, however, is not easily applicable to active perception tasks. This is because active perception tasks are closely related to sensory capabilities, and accurately simulating sensors (e.g. RGB, depth, touch, etc.) is difficult. Relative to prior work, we propose an RL method that efficiently learns active perception behavior on real robots while requiring minimal instrumentation (e.g. uncalibrated RGB cameras) at inference time.

To do this, we operate in an asymmetric training setting, exploiting privileged information [30] during training time to improve policy training [31, 32]. Privileged information approaches have been widely successful in solving partially observed tasks [33, 34] and have been deployed on real world robots with sim2real transfer [35–38]. As mentioned above however, it is difficult to perform sim-to-real transfer for active perception problems. Further, asymmetric RL approaches for sim2real [36, 37] are designed to exploit billions of privileged simulator state transitions [39], infeasible for privileged training in the real world, where we only have small amounts of potentially noisy privileged observations. We develop a new "asymmetric advantage weighted regression" RL algorithm that is more capable of learning efficiently in the real world, exploiting privileged additional sensors.

# 3  Asymmetric Reinforcement Learning in Active Perception POMDPs

Consider a robot tasked with finding a toy in a cluttered room using just its wrist camera, as seen in Figure 1. The toy's location is hidden to the robot, and it must scan the scene with its wrist camera in an efficient search path to find the toy quickly. These types of tasks where the robot has limited sensing but the reward and dynamics is dependent on some hidden environment state (e.g. toy location), are naturally modelled by partially observed Markov decision processes (POMDPs) [40].

A POMDP is represented by the tuple $(\mathcal{S}, \mathcal{A}, \mathcal{O}, T, R, E, P, \gamma)$ where $\mathcal{S}$ is the state space, $\mathcal{A}$ the action space, and $\mathcal{O}$ the observation space. The dynamics are described by the transition density $T(s_{t+1} \mid s_t, a_t)$, the reward density $R(r_t \mid s_t, a_t)$, the observation density $E(o_t \mid s_t)$, and the initial state density $P(s_0)$. For the search task of Figure 1, the state would include the robot position and toy location, while the observation would be the wrist camera view. The goal of policy synthesis in a POMDP is to find an optimal policy $\pi^*$ that maximizes the expected return $J(\pi) = \mathbb{E}^{\pi}[\sum_{t=0}^{\infty} \gamma^t r_t]$, where the discount factor $\gamma$ weights the importance of future rewards. In a POMDP, such a $\pi^*$ generally requires access to the complete history $h_t = (o_0, a_0, \ldots, o_t) \in \mathcal{H}$ of past observations and actions. This contrasts with an MDP, a special case of POMDP with $s_t = o_t$, in which the optimal policy depends only on the current state.

Back to POMDPs, it is usually impractical to learn a policy conditioned on the full history, since its input space would grow exponentially with time. Instead, it is common to consider an "agent state" $f: \mathcal{H} \to \mathcal{Z}$ that is recurrent in the sense that $z_t = f(h_t) = u(f(h_{t-1}), a_{t-1}, o_t)$, such as a sliding window. Then, the policy $\pi \in \Pi = \mathcal{Z} \to \Delta(\mathcal{A})$ must map from the agent state. Interestingly, when using an agent state and such policies, the POMDP can be transformed into an equivalent MDP whose state $(s_t, z_t)$ includes both the environment state and the agent state, with policies $\pi \in \Pi$ conditioned on the latter state only [41–44].

## 3.1  Background: Advantage Weighted Regression (AWR) for Markov Decision Processes

Advantage weighted regression (AWR) [13, 14] is a policy iteration algorithm for fully observed MDPs whose policy update objective is written as a behavior cloning loss, weighted by the estimated advantage of the transition. AWR is presented as a versatile algorithm that is able to leverage offline / off-policy data as well as on-policy data. More formally, at each iteration, AWR seeks to find a policy $\pi \colon \mathcal{S} \to \Delta(\mathcal{A})$ that maximizes the expected surrogate improvement, $\hat{\eta}(\pi) = \mathbb{E}_{s \sim d_\mu(s)} \mathbb{E}_{a \sim \pi(a|s)} A^\mu(s, a) \approx J(\pi) - J(\mu)$ with respect to a behavior policy $\mu$, under KL constraint $\mathbb{E}_{s \sim d_\mu(s)} [\mathrm{KL}(\pi(\cdot \mid s) \parallel \mu(\cdot \mid s)] \leq \varepsilon$. The behavior policy $\mu$ typically corresponds to the mixture

of all past policy iterates that generated the dataset of online interactions $\mathcal{D}_{\mathrm{on}}$. When relaxing the KL constraint with multiplier $\beta > 0$, optimizing this soft-constrained objective is equivalent to maximizing the final AWR objective:

$$\mathcal{L}_{\mathrm{AWR}}(\pi) = \underset{s \sim d_\mu(s)}{\mathbb{E}} \underset{a \sim \mu(a|s)}{\mathbb{E}} \left[ \exp \left( A^\mu(s, a)/\beta \right) \log \pi(a \mid s) \right]. \tag{1}$$

The original AWR algorithm [14] used either a return-based estimate or a TD($\lambda$) estimate of the advantage, by learning a value function with Monte Carlo estimation. Followup works [15, 16] used a critic-based estimate of the advantage by learning a Q-function with TD learning, which improves sample efficiency by better leveraging off-policy samples. We build on these latter works. For more detailed background on AWR and related approaches, see Appendix B.

## 3.2 The Need for Asymmetric Training in POMDPs

We now derive the AWR objective for POMDPs, showing why it requires asymmetry during training. We also show that the unprivileged value functions associated with a naive application of symmetric AWR cannot be learned by TD learning.

**Asymmetric and Symmetric AWR for POMDPs.** We aim to train a policy $\pi \colon \mathcal{Z} \to \Delta(\mathcal{A})$, conditioned on the agent state $z_t$ (equivalent to history $h_t = (o_1 \dots o_t)$) to maximize the return in POMDPs with an AWR-like objective. We consider the asymmetric learning paradigm in which the environment state $s$ available during training (offline or online) but not during policy deployment. We introduce the asymmetric AWR (AAWR) objective:

$$\mathcal{L}_{\mathrm{AAWR}}(\pi) = \underset{(s,z) \sim d_\mu(s,z)}{\mathbb{E}} \underset{a \sim \mu(a|z)}{\mathbb{E}} \left[ \exp \left( A^\mu(s, z, a)/\beta \right) \log \pi(a \mid z) \right] \tag{2}$$

where $A^\mu(s, z, a) = Q^\mu(s, z, a) - V^\mu(s, z)$ is the privileged advantage function, with $Q^\mu(s, z, a)$ and $V^\mu(s, z)$ the privileged critic and value functions, formally defined in Appendix C. See Figure 2 for a visual overview of the loss.

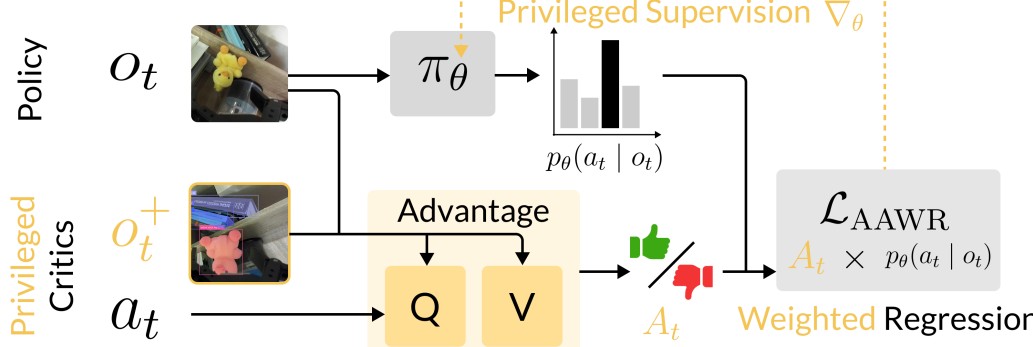

Figure 2: Top row: The policy receives the partial observation. Bottom row: Privileged observations or state, available only during training, are given to the critic networks to estimate the advantage. The advantage estimates are used as weights in the loss, providing privileged supervision to the policy.

If the environment state $s$ is unavailable during training, one natural strategy is to solely use agent state $z$ to estimate the advantage. We call this unprivileged variant the symmetric AWR (SAWR) objective, which is just Equation (2) with environment state $s$ removed from all terms.

Why is $\mathcal{L}_{\mathrm{AAWR}}$ the right objective to implement AWR for POMDPs? To show this, we start by observing that the original AWR objective was derived as constrained policy improvement in an MDP setting. To apply AWR to the POMDP setting with a policy $\pi \in \Pi = \mathcal{Z} \to \Delta(\mathcal{A})$, we can consider the equivalent MDP with state $(s, z)$, as discussed in Section 3. We then closely follow the original derivation in this MDP by additionally constraining the policy to be in $\Pi = \mathcal{Z} \to \Delta(\mathcal{A})$.

**Theorem 1** (Asymmetric Advantage Weighted Regression). For any POMDP and agent state $f \colon \mathcal{H} \rightarrow \mathcal{Z}$, the Lagrangian relaxation with Lagrangian multiplier $\beta > 0$ of the following constrained optimization problem,

$$\max_{\pi \in \Pi} \; \mathbb{E}_{(s,z) \sim d_\mu(s,z)} \; \mathbb{E}_{a \sim \pi(a|z)} \left[ A^\mu(s, z, a) \right] \tag{3}$$

$$\text{s.t.} \quad \mathbb{E}_{(s,z) \sim d_\mu(s,z)} \left[ \mathrm{KL}(\pi(\cdot \mid z) \parallel \mu(\cdot \mid z)) \right] \leq \varepsilon \tag{4}$$

is equivalent to the following optimization problem: $\max_{\pi \in \Pi} \mathcal{L}_{\mathrm{AAWR}}(\pi)$.

The proof is given in Appendix C and concludes the validity of the AAWR objective. In addition, we also show in Appendix D that optimizing the SAWR objective does not recover the correct solution, because an advantage estimator depending on agent state $z$ only is insufficient for estimating the advantage of the equivalent MDP, whose state is $(s, z)$. In the example in Figure 1, it is clear that a privileged advantage estimator with access to toy locations will better estimate success.

**Implementation Details.** To instantiate asymmetric advantage weighted regression, we train $V_\theta^\mu$ and critic $Q_\phi^\mu$ networks to compute the advantage, mirroring extensions [15, 16] of AWR that train critics to better leverage off-policy data instead of relying on MC returns. To train the networks, we choose IQL [45], a well known Q-learning algorithm known for its effectiveness in offline RL, offline-to-online RL finetuning [46] and real robot RL [47] tasks. The networks are trained using IQL's expectile regression objective, see Appendix A for details.

In our POMDPs, in the symmetric setting, the unprivileged advantage estimator would be $\hat{A}_{QV}^\mu(z_t, a_t) = Q_\phi^\mu(z_t, a_t) - V_\theta^\mu(z_t)$. In the asymmetric setting, the privileged advantage estimator would instead be $\hat{A}_{QV}^\mu(s_t, z_t, a_t) = Q_\phi^\mu(s_t, z_t, a_t) - V_\theta^\mu(s_t, z_t)$. In Appendix E, we show that the privileged value functions are the fixed point of the Bellman equations described by IQL's objective. In contrast, we show that the unprivileged value functions are not the fixed point of their corresponding Bellman equations, which further motivates the use of AAWR instead of SAWR.

We consider an asymmetric learning setting in which the state $s_t$ or privileged observations $o_t^p$ from additional sensors are available during offline / online training time, but not during policy deployment. The privileged critics take in either observation and state $(o_t, s_t)$, or the augmented observation $(o_t^+ = (o_t, o_t^p))$ while the policy only receives $o_t$.

| **Algorithm 1** AAWR Offline-to-Online Training |
| --- |
| **Require:** privileged $o^+$, partial $o$, $\pi(\cdot \mid o)$, $Q, V$ |
| 1: **for** $i = 1$ to $N_{\mathrm{off}}$ **do** |
| 2:      Update $Q, V$ using $\mathcal{D}_{\mathrm{off}}$ and IQL loss |
| 3:      Update $\pi$ using $\mathcal{D}_{\mathrm{off}}$ and Eq. 2 |
| 4: **for** $i = 1$ to $N_{\mathrm{on}}$ **do** |
| 5:      Collect $\{(o_t, o_t^+, a_t, r_t, o_{t+1}, o_{t+1}^+)\}_{t=1}^T$ with $\pi$ |
| 6:      $\mathcal{D}_{\mathrm{on}} \leftarrow \mathcal{D}_{\mathrm{on}} \cup \{(o_t, o_t^+, a_t, r_t, o_{t+1}, o_{t+1}^+)\}_{t=1}^T$ |
| 7:      Update $Q, V$ using $\mathcal{D}_{\mathrm{on}}, \mathcal{D}_{\mathrm{off}}$ and IQL loss |
| 8:      Update $\pi$ using $\mathcal{D}_{\mathrm{on}}, \mathcal{D}_{\mathrm{off}}$ and Eq. 2 |

| **Algorithm 2** Deployment |
| --- |
| **Require:** partial $o$, $\pi$ |
| 1: **for** $t = 1$ to $T$ **do** |
| 2:      $o \leftarrow \mathrm{env.step}(\pi(\cdot \mid o))$ |

Figure 3: Left: The policy is trained using privileged sensors on offline / online data. Right: After training, privileged sensors are no longer available and only the policy is deployed.

▶ **Privileged Training** We follow the offline-to-online RL paradigm [15, 45, 47–50] where the policy and value functions are first pre-trained on offline data using offline RL, and then are further fine-tuned with online interaction in the environment. Following lines 1-3 of Algorithm 1: Given the offline data $\mathcal{D}_{\mathrm{off}}$, we update $Q, V$ using the IQL objective and $\pi$ with the Equation (2) for $N_{\mathrm{off}}$ gradient steps. Next, in lines 4-8 of Algorithm 1: We execute the policy in the environment and store online transitions into buffer $\mathcal{D}_{\mathrm{on}}$. We sample an equal number of transitions from both buffers to form a batch, following best practice from prior work [47, 51]. Using the batch, we update $Q, V$ using the IQL objective and $\pi$ with the Equation (2).

▶ **Unprivileged Deployment**: During deployment, only partial observations $o$ remain available, which the policy uses to output actions, as seen in Algorithm 2.

# 4   Experiments

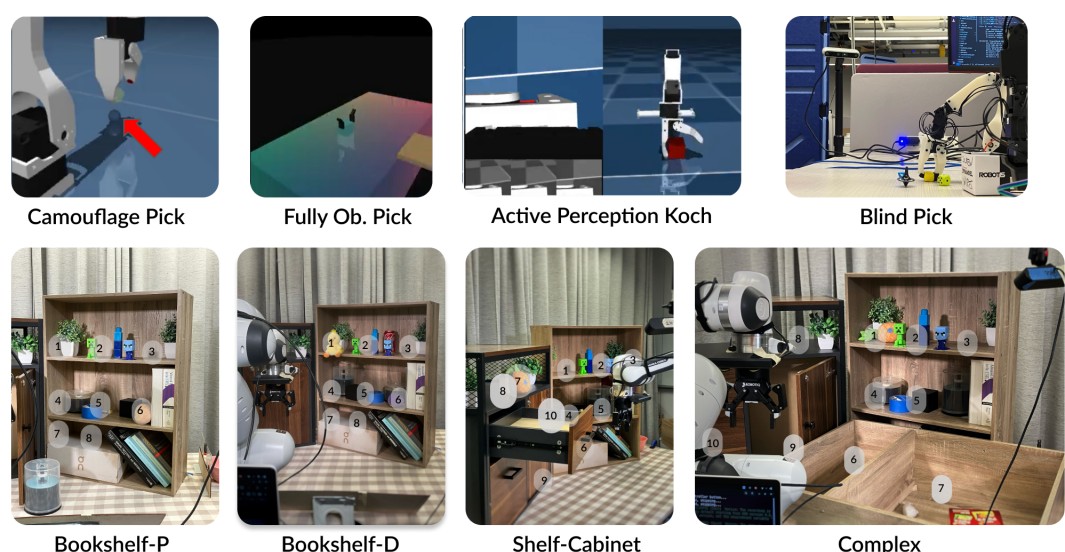

| Camouflage Pick | Fully Ob. Pick | Active Perception Koch | Blind Pick |
| Bookshelf-P | Bookshelf-D | Shelf-Cabinet | Complex |

Figure 4: We setup 8 different environments with diverse sensor setups and tasks to evaluate active perception behavior. Bottom row, we label the hiding spots for target objects.

In our experiments, we evaluate AAWR's ability to learn active and interactive perception behaviors in a variety of tasks. We aim to answer the following questions. Does AAWR learn active perception behaviors more efficiently than other approaches? Does AAWR work in both offline-to-online and purely offline training settings? Does active perception improve the ability of modern generalist policies to solve partially observed tasks?

## 4.1   Task setups

We evaluate AAWR in 8 different tasks, spanning both simulated and real world setups, see Figure 4 for images of all tasks. The tasks are grouped into simulated active perception tasks: **Camouflage Pick, Fully Obs. Pick, Active Perception Koch**, and real active/interactive perception tasks: **Blind Pick, Bookshelf-P, Bookshelf-D, Shelf-Cabinet, Complex**. In Table 1, we detail task properties like sensor setups, rewards, demonstrations, training budget, etc.

In the $\pi_0$ handoff tasks (**Blind Pick, Bookshelf-P, Bookshelf-D, Shelf-Cabinet, Complex**), we use three metrics to evaluate the search behavior. First, **Search** is a score of the policy behavior with the following rubric: 1) the target appears in wrist camera frame, 2) the policy approaches the target, and 3) the policy remains fixated on the target after 5 timesteps. Next, **Completion** is the rate at which $\pi_0$ successfully grabs the object after switching from the active perception policy. Finally, **Steps** is the amount of steps the policy takes to complete the task. See Appendices F to H for comprehensive descriptions of the tasks.

## 4.2   Baselines

We compare against symmetric advantage weighted regression (**AWR**) without privileged information. Its implementation is identical to that of AAWR, except for the inputs of the critic and value networks. Next, we compare against standard behavior cloning (**BC**), which performs imitation learning on the successful trajectories in the dataset.

## 4.3   Results

**Simulated Active Perception tasks.** Fig. 5 shows these results, and videos are in Supp and website. First, we compare against AWR and BC on two simulated active perception tasks with varied degrees

Table 1: The 8 tasks vary in embodiment, nature of partial observability, privileged sensors, reward, demo quantity and quality, and training budget.

| Task Platform | Target Obs. Privileged Obs. | Reward Demos | Offline Steps Online Steps | Description |
|---|---|---|---|---|
| Camouflage Pick Sim. Koch | Side Cam True Obj. Pos | Sparse 100 suboptimal | 20K 80K | Pick up barely visible object |
| Fully Obs. Pick Sim. xArm | Side Cam True Obj. Pos | Sparse 100 suboptimal | 20K 20K | Pick up fully visible object |
| AP Koch Sim. Koch | Wrist Cam True Obj. Pos | Sparse 100 suboptimal | 100K 900K | Locate then pick up object |
| Blind Pick Real Koch | Joints, Init Obj. Pos Obj. Pos Estimate | Dense 100 suboptimal | 20K 1.2K | Pick object from proprioception |
| Bookshelf-P Real Franka | Wrist Cam, Joints Bbox, Mask | Dense ~150 suboptimal | 100K 0 | Look for object & switch to $\pi_0$ |
| Bookshelf-D Real Franka | Wrist Cam, Joints Bbox, Mask | Dense ~100 suboptimal | 100K 0 | Look for object & switch to $\pi_0$ |
| Shelf-Cabinet Real Franka | Wrist Cam, Joints Bbox, Mask | Dense ~30 suboptimal | 100K 0 | Look for object & switch to $\pi_0$ |
| Complex Real Franka | Wrist Cam, Joints Bbox, Mask | Dense ~50 expert | 100K 0 | Look for object & switch to $\pi_0$ |

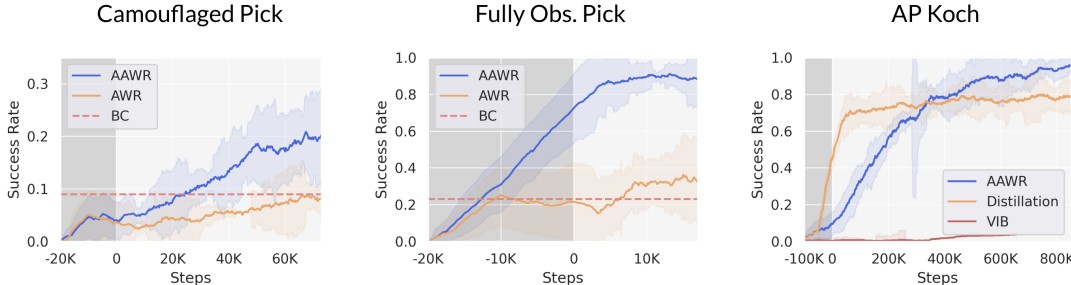

Figure 5: Evaluation curves for the simulated experiments, over 10 seeds per method. The shaded regions indicate the offline pretraining phase. AAWR outperforms baselines in all simulated tasks.

of observability, Camouflage Pick and Fully Obs. Pick. In both tasks, AAWR outperforms its non-privileged counterpart AWR, and BC in Camouflage Pick and Fully Obs. Pick, by approximately 2x and 3x respectively. While the gains from using privileged observations are in line with expectations for Camouflage Pick, where inferring the tiny marble from RGB is difficult, it is interesting to see gains even in a fully observable task, where the object position is always clearly inferable from vision. We hypothesize this is because the non-privileged critic needs to *learn* to extract the object position from pixels, whereas the privileged critic does not.

One natural question is: how does AAWR compare against other popular approaches that leverage privileged information? On the simulated **Active Perception Koch** task, we compare AAWR against **Distillation** [39], which first trains a privileged expert policy and then distills it into a partially observed policy. We also compare against a variational information bottleneck approach (**VIB**) [52], which gives the policy regularized access to privileged information.

As seen in Figure 5, only AAWR achieves 100% on the task by learning to scan the workspace, while other privileged baselines stagnate. Distillation gets high initial success but plateaus at 80%, learning a local maxima strategy of approaching the center of the workspace. This fails when the object is in the workspace corners, out of camera view. This behavior arises due to the privileged expert, unaware of the camera's limited field of view, that goes straight for the object. AAWR is a policy iteration algorithm that works with both offline and online data, which allows it to discover the

active perception behavior through online exploration, despite starting with suboptimal demos. VIB collapses during evaluation because privileged information is not available, even though it was trained to minimally use privileged information. See Appendix H for details and the website for videos.

**Real Interactive Perception task.** Here, we continue comparing to AWR and BC, this time including purely offline variants of AAWR and AWR, on the Blind Pick task in the real world. As seen in Table 2, both offline and online variants of AAWR outperform its unprivileged counterparts, and BC. Among the offline methods, BC performs worst, exhibiting jerky and inaccurate movements. Both offline AWR and offline AAWR demonstrate better approaching and picking behavior, but offline AWR missed grasps and released the block frequently.

Table 2: Koch Interactive Perception.

| Method | Grasp % | Pick % |
|---|---|---|
| BC | 47 | 41 |
| Off. AWR | 65 | 62 |
| On. AWR | 71 | 55 |
| Off. AAWR (ours) | 88 | 71 |
| On. AAWR (ours) | **94** | **89** |

Offline AAWR demonstrates more suboptimal behavior, such as releasing the candy after grasping. We observed that after online finetuning, the suboptimal behavior of offline AAWR is reduced, and online AAWR demonstrates the most consistent and robust open-loop picking behavior. Online AAWR reliably places its gripper over the object for grasping. In cases when the object slips from its grasp, the policy attempts to regrasp at the original location. See videos on the website.

Table 3: In the active perception handoff tasks, AAWR consistently outperforms baselines in terms of search behavior, completion, and speed. Bold = best, underline = second best.

| Method | Bookshelf-P | | | Bookshelf-D | | | Shelf-Cabinet | | | Complex | | |
|---|---|---|---|---|---|---|---|---|---|---|---|---|
| | Search ↑ | Completion ↑ | Steps ↓ | Search ↑ | Completion ↑ | Steps ↓ | Search ↑ | Completion ↑ | Steps ↓ | Search ↑ | Completion ↑ | Steps ↓ |
| **AAWR** | **92.4** | **44.4** | 36.6 | 81.3 | **44.4** | 26.9 | **78.2** | 40.0 | 46.3 | 73.2 | **50.0** | **43.0** |
| AWR | 79.6 | 0.0 | **34.0** | 62.6 | 16.7 | 30.2 | 52.3 | 10.0 | **38.0** | 33.2 | 40.0 | 67.0 |
| BC | 29.9 | 20.0 | 84.0 | 47.7 | 16.7 | **22.5** | 28.1 | 15.0 | 125.0 | 31.5 | 15.0 | 77.0 |
| $\pi_0$ | 11.0 | 16.7 | 263.3 | 66.7 | 33.3 | 229.7 | 10.0 | 10.0 | 280.0 | 29.6 | 20.0 | 252.5 |
| Exhaustive | 64.2 | 44.0 | 105.4 | **96.0** | 22.2 | 106.7 | 52.8 | **45.0** | 183.0 | **78.2** | 30.0 | 297.0 |
| VLM+$\pi_0$ | 31.4 | 27.8 | 322.3 | 33.2 | 16.7 | 281.8 | 28.2 | 15.0 | 382.0 | 14.8 | 10.0 | 374.7 |

**Handholding Foundation VLA Policies for Real Active Perception tasks.** We find that $\pi_0$, a generalist foundation policy for manipulation tasks, is not good at searching tasks such as finding target objects in a cluttered scene from just a wrist camera (see in Figure 4). $\pi_0$ is fundamentally limited by not having memory, which hampers its ability to handle POMDPs. We now evaluate our approach's ability to generate helper policies that condition on history (see Appendix F) to handhold $\pi_0$ policies up to a configuration from which they could reasonably succeed. To achieve this, we propose a switching framework for the policy where an active perception policy is first run, and an object detector periodically checks the wrist image for the target object. Once the object is detected across two consecutive intervals, the robot switches to $\pi_0$ to grasp the object.

We set up four realistic active perception tasks where the robot must search through a cluttered scene to find a target object and grasp it up (see Figure 4). In the **Bookshelf-P** and **Bookshelf-D** tasks, we place a target object (either a toy pineapple or duck) on one of the three shelves, requiring the robot to scan the shelves both vertically and horizontally and stop at good viewpoint. In the **Shelf-Cabinet** task, we add an additional cabinet with drawers and hiding spots near its top and inside cabinet door, increasing the complexity of the search. Finally, in **Complex** task, we add an additional shelf on the floor, whose shelves are completely occluded from all side camera viewpoints.

We compare against several baselines. **Exhaustive** is an engineered controller that methodically goes through all hiding spots in the scene, but is slow due to its heuristic pathfinding. Next, we again compare to unprivileged AWR and BC as helper policies. We compare against $\pi_0$ itself, as well as a **VLM+$\pi_0$** variant that queries the Gemini-2.5 VLM [53] to generate language commands that are executed by $\pi_0$, similar to the Hierarchical VLM-VLA baseline proposed in [54]. As mentioned earlier in Section 4.1, we use the **Search** metric to judge the searching behavior, **Completion** metric to measure how useful is the handoff for $\pi_0$, and **Steps** for speed. See Appendix F for more details.

For the **Bookshelf-P, Bookshelf-D** tasks, we collected 250 demonstrations split among four demonstrators with varying qualities, resulting in a suboptimal dataset of about 50% $\pi_0$ success rate. For

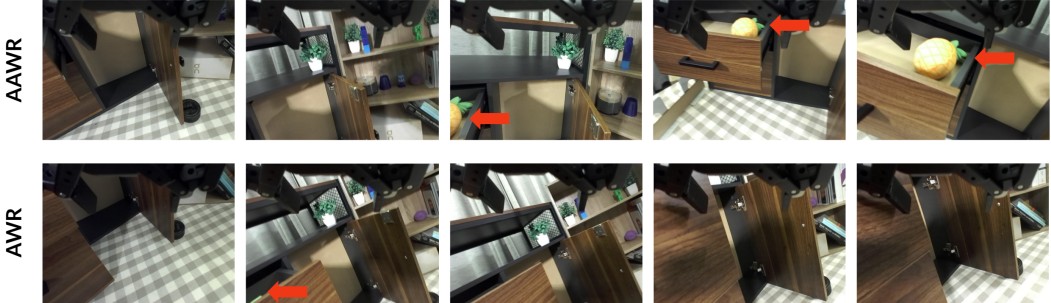

Figure 6: Example rollouts in the cabinet shelf task. AAWR explores the top left of the cabinet and locates the target object (red arrow) while AWR only finds a small glimpse before drifting away.

**Shelf-Cabinet, Complex**, we collected smaller but higher quality datasets of 35 trajectories with 74% success rate and 50 trajectories with 94% success rate respectively. We perform offline training of AAWR, AWR, and BC on the same demonstration dataset, and for the same amount of gradient steps, before evaluating in the real world.

As seen in Table 4, AAWR consistently outperforms baselines in all metrics, learning sensible active perception behavior to aid a generalist policy. AAWR always outperforms non privileged AWR and BC, thus validating the usefulness of privileged information and the use of offline RL over supervised learning. We find that $\pi_0$ and VLM+$\pi_0$ both search poorly - they tend to take inefficient movements and fail to track the object. Exhaustive has decent search and completion rates, but takes much longer than AAWR. In fact, when normalized for time taken, AAWR scores 2-8 times higher in Search and Completion metrics than Exhaustive. See Table 5 for the time-normalized relative metrics.

In the **Bookshelf** tasks, AAWR first learns to zoom out of the scene to see multiple shelves, then scans from bottom to up, and then approaches the target object once located. The AWR and BC baselines follow a relatively fixed search path that approaches the shelf, but the policies failed to efficiently scan the shelves. Even if the target object appears in the frame, the policies do not fixate on the object, reducing their search score and $\pi_0$ success rate. In the **Shelf-Cabinet** task, AAWR searches through the right bookshelf, before moving to the left cabinet. Both AWR and BC do not thoroughly search the scene, preventing them from finding objects placed in the left cabinet's drawer. In **Complex**, AAWR searches the bottom shelf, the right shelf, and then the left cabinet (see Figure 1). See Figure 6 and the website for examples.

## 5   Conclusion and Limitations

We aim to train active perception policies in the real world to allow robots to overcome their sensory limitations, a useful problem for which current approaches have shown limited success. We propose asymmetric advantage weighted regression (AAWR), a simple weighted behavior cloning technique that leverages privileged observations during training to efficiently train active perception policies. We provide a theoretical justification for AAWR, by deriving the validity of the AAWR objective for POMDP as opposed to its symmetric counterpart. Then, we show that AAWR successfully learns active and interactive perception behaviors in 8 different simulated and real world tasks.

Despite our promising results, there is much room for improvement. Instead of switching between policies to execute active perception behaviors, AAWR could be used to directly fine-tune generalist foundation policies. Other forms of privileged information could be considered, such as foundation model outputs. Instead of using prespecified privileged information, useful features from the additional information could also be explicitly selected through representation learning. Because this work overcomes limitations of existing methods for active perceptions tasks, it would be worth exploring the scalability of AAWR to tasks with longer horizons where information-gathering challenges compound. Finally, AAWR could be applied to many other partially observed robotic (or other) tasks.

## Acknowledgments

This work was supported by the DARPA TIAMAT HR0011249042, NSF CAREER 2239301, ONR N00014-22-1-2677, and NSF SLES 2331783 grants. It used computing resources from the National Artificial Intelligence Research Resource Pilot (NAIRR 240077). Gaspard Lambrechts is a postdoctoral researcher of the *Fund for Scientific Research* (FNRS) from the *Wallonia-Brussels Federation* in Belgium. The authors would like to thank the GRASP lab, anonymous NeurIPS reviewers, Antonio Loquercio, Kostas Daniilidis, Lei Zhang, Tianyou Wang, Siyu Li and Jiahui Lei for constructive feedback and support. The authors thank James Springer, Will Liang, Johnny Wang, Sam Wang and Tau Robotics for robot support.

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

## A  AAWR Implementation Details

Instead of the standard policy evaluation of the AWR algorithm [14–16], we use implicit Q-learning (IQL) [45] and its optimistic policy evaluation, for its effectiveness in offline, and offline-to-online [46] and real robots [47] tasks. The IQL algorithm learns both a value function $V_\theta^\mu$ and a critic $Q_\phi^\mu$. Extending IQL to our POMDPs, in the symmetric setting, the unprivileged advantage estimator would be $\hat{A}_{QV}^\mu(z_t, a_t) = Q_\phi^\mu(z_t, a_t) - V_\theta^\mu(z_t)$. In the asymmetric setting, the privileged advantage estimator would instead be $\hat{A}_{QV}^\mu(s_t, z_t, a_t) = Q_\phi^\mu(s_t, z_t, a_t) - V_\theta^\mu(s_t, z_t)$.

The privileged Q-function is trained using the 1-step TD error.

$$\mathcal{L}_Q(\phi) = \mathop{\mathbb{E}}_{(s_t, z_t, r_t, s_{t+1}, z_{t+1}) \sim \mathcal{D}} \left[ (r_t + \gamma V_\theta^\mu(s_{t+1}, z_{t+1}) - Q_\phi^\mu(s_t, z_t, a_t))^2 \right] \tag{5}$$

The privileged value function is trained to conservatively approximate the maximization $\max_a Q_\phi^\mu(s, z, a)$ using an asymmetric $L_2$ loss (expectile regression):

$$\mathcal{L}_V(\theta) = \mathop{\mathbb{E}}_{(s_t, z_t, a_t) \sim \mathcal{D}} \left[ |\tau - \mathbb{1}_{\{Q_\phi^\mu(s_t, z_t, a_t) - V_\theta^\mu(s_t, z_t) < 0\}}| (Q_\phi^\mu(s_t, z_t, a_t) - V_\theta^\mu(s_t, z_t))^2 \right] \tag{6}$$

where $\tau \in (0, 1)$ is the expectile. As $\tau \to 0$, the loss increasingly penalizes overestimates of $V$. The unprivileged value functions are trained analogously.

Note that when $\tau = 0.5$, it corresponds to the standard 1-step TD update. In Appendix E, we show that the privileged value functions are the fixed point of the Bellman equations described by Eq. 5 and Eq. 6. In contrast, we show that the unprivileged value functions are not the fixed point of their corresponding Bellman equations, which further motivates the use of AAWR instead of SAWR.

---

**Algorithm 3** AAWR Offline-to-Online
___
**Require:** policy $\pi$, critics $Q, V$, buffers $\mathcal{D}_{\text{off}}, \mathcal{D}_{\text{on}}$
1: **for** $i = 1$ to $N_{\text{off}}$ **do**
2:     Update $Q, V$ using $\mathcal{D}_{\text{off}}$ and Eq. 5 and Eq. 6
3:     Update $\pi$ using $\mathcal{D}_{\text{off}}$ and Eq. 2
4: **for** $i = 1$ to $N_{\text{on}}$ **do**
5:     Collect $\{(o_t, o_t^+, a_t, r_t, o_{t+1}, o_{t+1}^+)\}_{t=1}^T$ with $\pi$
6:     $\mathcal{D}_{\text{on}} \leftarrow \mathcal{D}_{\text{on}} \cup \{(o_t, o_t^+, a_t, r_t, o_{t+1}, o_{t+1}^+)\}_{t=1}^T$
7:     Update $Q, V$ using $\mathcal{D}_{\text{on}}, \mathcal{D}_{\text{off}}$ and Eq. 5, Eq. 6
8:     Update $\pi$ using $\mathcal{D}_{\text{on}}, \mathcal{D}_{\text{off}}$ and Eq. 2

---

We train the models in an offline-to-online manner. Following lines 1-3 of Algorithm 3: during the offline stage, $Q, V, \pi$ are updated $\mathcal{D}_{\text{off}}$ for $N_{\text{off}}$ gradient steps on the offline dataset. After offline training, on-policy trajectories are collected by executing the target policy in the environment. These trajectories are added to the online buffer $\mathcal{D}_{\text{on}}$. Following [47], we use symmetric sampling where 50% of samples are from $\mathcal{D}_{\text{off}}$ and the other 50% are from $\mathcal{D}_{\text{on}}$.

## B  Advantage Weighted Regression Estimators

The original AWR algorithm [14] used a return-based estimate of the advantage $\hat{A}_V^\mu(s_t, a_t) = \sum_{k=0}^\infty \gamma^k r_{t+k} - \hat{V}^\mu(s_t)$ where $\hat{V}^\mu(s_t)$ is an approximation of the value function, learned by minimization of,

$$\mathcal{L}_{\text{MC}}(\hat{V}) = \mathop{\mathbb{E}}_{s_t \sim d_\mu(s_t)} \left[ \left( \sum_{k=0}^\infty \gamma^k r_{t+k} - \hat{V}(s_t) \right)^2 \right]. \tag{7}$$

Future works have instead used a critic-based estimate of the advantage $\hat{A}_Q^\mu(s_t, a_t) = Q_\phi^\mu(s_t, a_t) - \mathbb{E}_{a \sim \pi(a|s_t)} Q_\phi^\mu(s_t, a)$ [15, 16] where $Q_\phi^\mu(s_t, a_t)$ is an approximation of the critic function, learned by minimization of,

$$\mathcal{L}_{\text{TD}}(\hat{Q}) = \mathop{\mathbb{E}}_{s_t \sim d_\mu(s_t)} \mathop{\mathbb{E}}_{a_t \sim \pi(a_t|s_t)} \left[ (r_t + \gamma \hat{Q}'(s_{t+1}, a_{t+1}) - \hat{Q}(s_t, a_t))^2 \right]. \tag{8}$$

The typical learning procedure of AWR is summarized in Algorithm 4.

---
**Algorithm 4** Advantage Weighted Regression
---
**Require:** policy $\pi$, critic $V$, buffer $\mathcal{D}_{\text{on}}$
  1: **for** $i = 1$ to $N_{\text{on}}$ **do**
  2:      Collect $\{(s_t, a_t, r_t, s_{t+1})\}_{t=1}^{T}$ with $\pi$
  3:      $\mathcal{D}_{\text{on}} \leftarrow \mathcal{D}_{\text{on}} \cup \{(s_t, a_t, r_t, s_{t+1})\}_{t=1}^{T}$
  4:      Update $V$ using $\mathcal{D}_{\text{on}}$ and minimizing Eq. 7
  5:      Update $\pi$ using $\mathcal{D}_{\text{on}}$ and maximizing Eq. 1
---

## C   Asymmetric Advantage Weighted Regression Derivation

In this section, we derive the AWR objective for POMDPs, which results in the AAWR objective. Since we consider a POMDP and an agent state $f\colon \mathcal{H} \rightarrow \mathcal{Z}$, we can consider the equivalent environment-agent state MDP [42–44], whose state is $(s, z)$, and restrain the class of fully observable policies $\Pi^+ = \mathcal{S} \times \mathcal{Z} \rightarrow \Delta(\mathcal{A})$ for this MDP to the agent-state policies,

$$\Pi^- = \left\{ \pi^+ \in \Pi^+ \mid \exists \pi \in \Pi : \pi^+(a \mid s, z) = \pi(a \mid z), \forall s \in \mathcal{S},\ \forall z \in \mathcal{Z},\ \forall a \in \mathcal{A} \right\}. \quad (9)$$

Since $\Pi^- \subseteq \Pi^+$, we have $\pi^- \in \Pi^+$ and we can derive the AWR objective using this restricted set of policies following similar steps as Peng et al. [14]. In the following, we use $\pi \in \Pi$ to denote the partially observable policy $\pi(a \mid z)$ corresponding to $\pi^- \in \Pi^-$ with $\pi^-(a \mid s, z) = \pi(a \mid z)$. Before deriving the AWR objective for the equivalent environment-agent state MDP, let us define the (normalized) discounted visitation measure of a policy $\pi \in \Pi$ in this MDP as:

$$d_\pi(s, z) = (1 - \gamma) \sum_{t=0}^{\infty} \gamma^t p(s_t = s, z_t = z). \quad (10)$$

In the following, we denote the current policy $\pi_k^-$ with $\mu^-$, and its corresponding partially observable policy with $\mu$ Since we work in the environment-agent state MDP, we consider the usual value functions definitions, where the state is $(s, z)$,

$$Q^{\mu^-}(s, z, a) = \mathbb{E}^{\mu^-}\left[ \sum_{t=0}^{\infty} \gamma^t r_t \,\middle|\, s_0 = s, z_0 = z, a_0 = a \right] \quad (11)$$

$$V^{\mu^-}(s, z) = \mathbb{E}^{\mu^-}\left[ \sum_{t=0}^{\infty} \gamma^t r_t \,\middle|\, s_0 = s, z_0 = z \right]. \quad (12)$$

Now, let us derive an objective to improve the policy. We want to maximize the policy improvement,

$$\eta(\pi^-) = J(\pi^-) - J(\mu^-). \quad (13)$$

The policy improvement is related to the advantage function of $\mu^-$ [55],

$$\eta(\pi^-) = \mathbb{E}^{\pi^-}\left[\sum_{t=0}^{\infty}\gamma^t r_t\right] - \mathbb{E}^{\mu^-}\left[\sum_{t=0}^{\infty}\gamma^t r_t\right] \tag{14}$$

$$= \mathbb{E}^{\pi^-}\left[\sum_{t=0}^{\infty}\gamma^t r_t\right] - \mathbb{E}\left[V^{\mu^-}(s_0, z_0)\right] \tag{15}$$

$$= \mathbb{E}^{\pi^-}\left[\sum_{t=0}^{\infty}\gamma^t r_t - V^{\mu^-}(s_0, z_0)\right] \tag{16}$$

$$= \mathbb{E}^{\pi^-}\left[\sum_{t=0}^{\infty}\gamma^t(r_t + \gamma V^{\mu^-}(s_{t+1}, z_{t+1}) - V^{\mu^-}(s_t, z_t)) + V^{\mu^-}(s_0, z_0) - V^{\mu^-}(s_0, z_0)\right] \tag{17}$$

$$= \mathbb{E}^{\pi^-}\left[\sum_{t=0}^{\infty}\gamma^t(r_t + \gamma V^{\mu^-}(s_{t+1}, z_{t+1}) - V^{\mu^-}(s_t, z_t))\right] \tag{18}$$

$$= \mathbb{E}^{\pi^-}\left[\sum_{t=0}^{\infty}\gamma^t A^{\mu^-}(s_t, z_t, a_t)\right] \tag{19}$$

$$= \sum_{t=0}^{\infty}\gamma^t \mathbb{E}^{\pi^-}\left[A^{\mu^-}(s_t, z_t, a_t)\right] \tag{20}$$

$$= (1 - \gamma)\mathop{\mathbb{E}}_{(s,z)\sim d_\pi(s,z)}\mathop{\mathbb{E}}_{a\sim\pi(a|z)}\left[A^{\mu^-}(s, z, a)\right] \tag{21}$$

where $A^{\mu^-}(s, z, a) = Q^{\mu^-}(s, z, a) - V^{\mu^-}(s, z)$. The objective $\eta(\pi^-)$ may be inefficient to optimize, due to the dependence of the expectation on $\pi^-$ through $d_\pi$. Instead, we thus choose to optimize an off-policy surrogate objective where the samples are generated from policy $\mu$,

$$\hat{\eta}(\pi^-) = (1 - \gamma)\mathop{\mathbb{E}}_{(s,z)\sim d_\mu(s,z)}\mathop{\mathbb{E}}_{a\sim\pi(a|z)}\left[A^{\mu^-}(s, z, a)\right] \tag{22}$$

In practice, we thus seek to approximately solve the following constrained optimization problem at each iteration,

$$\pi^-_{k+1} \in \arg\max_{\pi^-\in\Pi^-}\mathop{\mathbb{E}}_{(s,z)\sim d_\mu(s,z)}\mathop{\mathbb{E}}_{a\sim\pi(a|z)}\left[A^{\mu^-}(s, z, a))\right] \tag{23}$$

$$\text{s.t. } \mathrm{KL}(\pi^-(\cdot \mid s, z) \,\|\, \mu^-(\cdot \mid s, z)) \leq \epsilon \tag{24}$$

Now that we have identified the desired constrained optimization problem, let us prove Theorem 1, which states that its relaxation corresponds to the AAWR objective.

**Theorem 1** (Asymmetric Advantage Weighted Regression). For any POMDP and agent state $f\colon \mathcal{H} \to \mathcal{Z}$, the Lagrangian relaxation with Lagrangian multiplier $\beta > 0$ of the following constrained optimization problem,

$$\max_{\pi\in\Pi}\mathop{\mathbb{E}}_{(s,z)\sim d_\mu(s,z)}\mathop{\mathbb{E}}_{a\sim\pi(a|z)}\left[A^{\mu}(s, z, a)\right] \tag{3}$$

$$\text{s.t. } \mathop{\mathbb{E}}_{(s,z)\sim d_\mu(s,z)}\left[\mathrm{KL}(\pi(\cdot \mid z) \,\|\, \mu(\cdot \mid z)\right] \leq \varepsilon \tag{4}$$

is equivalent to the following optimization problem: $\max_{\pi\in\Pi}\mathcal{L}_{\mathrm{AAWR}}(\pi)$.

*Proof.* Starting from Eq. 23, we note that two additional constraints are hidden in $\pi^- \in \Pi^-$. The first one is $\int_{a\in\mathcal{A}}\pi(a \mid s, z) = 1$, $\forall s \in \mathcal{S}$, $\forall z \in \mathcal{Z}$. The second one is $\pi^- \in \Pi^- \subseteq \Pi^+$, or equivalently, $\pi^-(a \mid s_1, z) = \pi^-(a \mid s_2, z)$, $\forall s_1, s_2 \in \mathcal{S}$, $\forall z \in \mathcal{Z}$, $\forall a \in \mathcal{A}$.

Following similar steps as Peng et al. [14], by relaxing the KL constraint in a Lagrangian multiplier with multiplier $\beta$,

$$\pi^*(a \mid s, z) \propto \mu^-(a \mid s, z) \exp\left(\frac{1}{\beta} A^{\mu^-}(s, z, a)\right) \tag{25}$$

$$\propto \mu(a \mid z) \exp\left(\frac{1}{\beta} A^{\mu^-}(s, z, a)\right) \tag{26}$$

We now substitute back the two additional constraints as additional constraints, so that we project the solution on the manifold of policies $\Pi^-$. By minimizing the KL divergence $\mathbb{E}_{(s,z)\sim d_\mu(s,z)}\left[\text{KL}(\pi^*(\cdot \mid s, z) \parallel \pi(\cdot \mid z))\right]$ to that target, we obtain

$$\pi_{k+1} = \arg\max_{\pi \in \Pi} \mathbb{E}_{(s,z)\sim d_\mu(s,z)} \mathbb{E}_{a\sim\mu(a|z)} \left[\log \pi(a \mid z) \exp\left(\frac{1}{\beta} A^{\mu^-}(s, z, a)\right)\right] \tag{27}$$

This concludes the proof, for $\mu = \pi_k$. □

With the additional constraint that we use a parametrized policy $\pi_\theta \in \Pi_\Theta$, we obtain

$$\theta_{k+1} = \arg\max_{\theta \in \Theta} \mathbb{E}_{(s,z)\sim d_\mu(s,z)} \mathbb{E}_{a\sim\mu(a|z)} \left[\log \pi_\theta(a \mid z) \exp\left(\frac{1}{\beta} A^{\mu^-}(s, z, a)\right)\right] \tag{28}$$

Eq. 27 corresponds to the AAWR objective, and Eq. 28 is the AAWR objective in the context of parametrized function approximators.

## D  Problem with Symmetric Advantage Weighted Regression

While the asymmetric value functions followed the classical definitions in the environment-agent state MDP, the symmetric value functions are not standard because $z$ is not a Markovian state. We select the following definition: $A^\mu(z, a) = Q^\mu(z, a) - V^\mu(z)$ with,

$$Q^{\mu^-}(z, a) = \mathbb{E}_{s\sim d_{\mu^-}(s|z)} \left[\mathbb{E}^{\mu^-} \left[\sum_{t=0}^\infty \gamma^t r_t \middle| s_0 = s, z_0 = z, a_0 = a\right]\right] \tag{29}$$

$$V^{\mu^-}(z) = \mathbb{E}_{s\sim d_{\mu^-}(s|z)} \left[\mathbb{E}^{\mu^-} \left[\sum_{t=0}^\infty \gamma^t r_t \middle| s_0 = s, z_0 = z\right]\right] \tag{30}$$

By definition, this choice provides unbiased symmetric value functions under the distribution $d_{\mu^-}(s, z)$ induced by the current policy $\mu^-$.

$$Q^{\mu^-}(z, a) = \mathbb{E}_{s\sim d_{\mu^-}(s|z)} \left[Q^{\mu^-}(s, z, a)\right] \tag{31}$$

$$V^{\mu^-}(z) = \mathbb{E}_{s\sim d_{\mu^-}(s|z)} \left[V^{\mu^-}(s, z)\right] \tag{32}$$

Let us now prove Theorem 2, which proves that the symmetric AWR (SAWR) objective is different from the asymmetric AWR (AAWR) objective.

**Theorem 2** (Symmetric Advantage Weighted Regression). In general, for a POMDP and an agent state $f \colon \mathcal{H} \to \mathcal{Z}$, we have $\arg\max_{\pi \in \Pi} \mathcal{L}_{\text{SAWR}} \neq \arg\max_{\pi \in \Pi} \mathcal{L}_{\text{AAWR}}$.

*Proof.* Combining Eq. 29 and Eq. 30, we have,

$$A^{\mu^-}(z, a) = \mathbb{E}_{s\sim d_{\mu^-}(s|z)} \left[A^{\mu^-}(s, z, a)\right] \tag{33}$$

Now, it is straightforward to see that the SAWR objective,

$$\mathcal{L}_{\text{SAWR}}(\pi) = \mathbb{E}_{(s,z)\sim d_\mu(s,z)} \mathbb{E}_{a\sim\mu(a|z)} \left[\exp\left(A^{\mu^-}(z, a)/\beta\right) \log \pi(a \mid z)\right] \tag{34}$$

$$= \mathbb{E}_{(s,z)\sim d_\mu(s,z)} \mathbb{E}_{a\sim\mu(a|z)} \left[\exp\left(\mathbb{E}_{s\sim d_{\mu^-}(s|z)} \left[A^{\mu^-}(s, z, a)\right]/\beta\right) \log \pi(a \mid z)\right] \tag{35}$$

does not correspond to the AAWR objective,

$$\mathcal{L}_{\text{AAWR}}(\pi) = \underset{(s,z)\sim d_\mu(s,z)}{\mathbb{E}} \underset{a\sim\mu(a|z)}{\mathbb{E}} \left[ \exp\left( A^{\mu^-}(s,z,a) \right) \log \pi(a \mid z) \right]. \tag{36}$$

Indeed, we can apply Jensen's strict inequality over the the strictly convex exponential function, under the assumption that the distribution $d_{\mu^-}(s \mid z)$ is nondegenerate. □

# E  Problem with Symmetric Temporal Difference Learning

Let us consider the asymmetric Bellman equations for the environment-agent state MDP,

$$\tilde{Q}^{\mu^-}(s,z,a) = \mathbb{E}^{\mu^-}\left[ r_t + \gamma \tilde{Q}^{\mu^-}(s_{t+1}, z_{t+1}, a_{t+1}) \Big| s_t = s, z_t = z, a_t = a \right]. \tag{37}$$

Since the underlying Bellman operator is $\gamma$-contractive, these equations have a unique solution $\tilde{Q}^{\mu^-}$. Because the environment and agent states form a Markovian variable $(s,z)$, by definition of $Q^{\mu^-}$, we have $\tilde{Q}^{\mu^-} = Q^{\mu^-}$. As a consequence, we also have $\tilde{V}^{\mu^-} = V^{\mu^-}$ where $\tilde{V}^{\mu^-}$ is the unique fixed point of its analogous Bellman operator.

Let us now consider the symmetric Bellman equations for the environment-agent state MDP,

$$\tilde{Q}^{\mu^-}(z,a) = \underset{s'\sim d_{\mu^-}(s'|z)}{\mathbb{E}} \left[ \mathbb{E}^{\mu^-}\left[ r_t + \gamma \tilde{Q}^{\mu^-}(z_{t+1}, a_{t+1}) \Big| s_t = s', z_t = z, a_t = a \right] \right]. \tag{38}$$

It is interesting to note that, by bootstrapping with $\tilde{Q}^{\mu^-}$, this Q-function considers the distribution of the state $(s_{t+1}, z_{t+1} \mid s_t, z_t, a_t)$ from the second timestep to be $p(z_{t+1} \mid s_t, z_t, a_t)d_{\mu^-}(s_{t+1}|z_{t+1})$ instead of the true distribution $p(s_{t+1}, z_{t+1} \mid s_t, z_t, a_t)$. As a result, by telescoping, we obtain,

$$\tilde{Q}^{\mu^-}(z,a) = \underset{s\sim d_{\mu^-}(s|z)}{\mathbb{E}} \left[ \sum_{t=0}^\infty \gamma^t \mathbb{E}^{\mu^-}\left[ \underset{s'_t\sim d_{\mu^-}(s'_t|z_t)}{\mathbb{E}} r(s'_t, z_t, a_t) \Big| s_0 = s, z_0 = z, a_0 = a \right] \right] \tag{39}$$

where $r(s,z,a) = \mathbb{E}[r_t \mid s_t = s, z_t = z, a_t = a]$. It contrasts with the true Q-function, which writes,

$$Q^{\mu^-}(s,z,a) = \mathbb{E}^{\mu^-}\left[ \sum_{t=0}^\infty \gamma^t r_t \Big| s_0 = s, z_0 = z, a_0 = a \right] \tag{40}$$

$$= \underset{s\sim d_{\mu^-}(s|z)}{\mathbb{E}} \left[ \sum_{t=0}^\infty \gamma^t \mathbb{E}^{\mu^-}\left[ r(s_t, z_t, a_t) | s_0 = s, z_0 = z, a_0 = a \right] \right] \tag{41}$$

It can be concluded that the unprivileged fixed point $\tilde{Q}^{\mu^-}$ and the unprivileged Q-function $Q^{\mu^-}$ can be different, as soon as the distribution $p(s_t, z_t, a_t \mid s_0, z_0, a_0)$ is different from $p(z_t, a_t \mid s_0, z_0, a_0)d^{\mu^-}(s_t \mid z_t)$ at any timestep $t$.

# F  Active Perception Experimental Details

## F.1  Task Definition

In these tasks, the robot must find objects placed out of view in cluttered environments. Similar to how a human may try to find ingredients in a messy kitchen, the robot must move its camera around occluders, zoom into hard to see spots (i.e. behind drawers), and zoom out to increase its overall view of the scene.

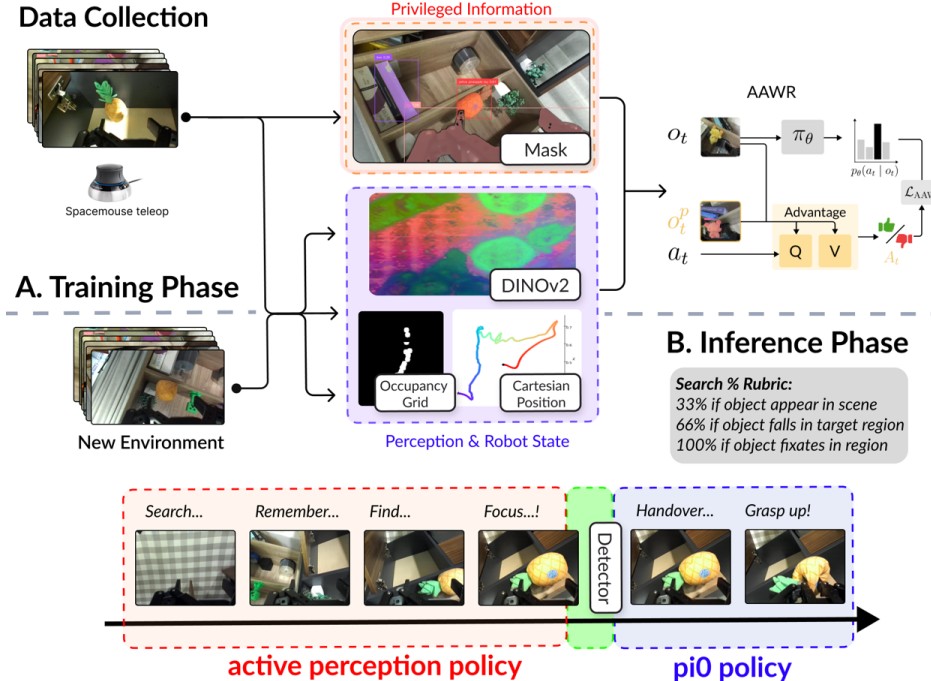

Figure 7: **Franka Robot Search Diagram**: A visual overview of the Franka Active Perception experiments. Top half: Offline trajectories are collected through teleoperation collecting privileged masks and unprivileged wrist image and history features. The policy is trained using AAWR. Bottom half: During evaluation, only the unprivileged search policy is run to perform active perception to find the object, before switching to the generalist policy. The policy's search behavior is graded with the rubric.

In this experiment, we train a "helper" active perception policy that searches the scene for the target object, and once located, hands off control to a generalist VLA policy to pick up the object. This addresses a weakness of such generalist policies - that they are typically trained in fully observed situations, and do not generalize to partial observations.

We set up four tasks, ordered in increasing complexity, where the robot must either find a toy *pineapple* or *duck*. Objects and visual distractors are randomly placed at the beginning of each episode, the bookshelf and cabinet are placed in same position.

- **Bookshelf-P** and **Bookshelf-D**: The robot must find either the pineapple or duck in a three-tier vertical bookshelf with multiple visual distractors (Fig. Figure 9).

- **Shelf-Cabinet**: We make the scene more difficult by adding a cabinet on the left side. The cabinet creates additional hiding spots on its top drawer and shelf over the drawer.

- **Complex**: In addition to the bookshelf and cabinet, we add a horizontal bookshelf. There are several completely occluded spots, such as the bottom cabinet drawer and objects placed in the horizontal bookshelf.

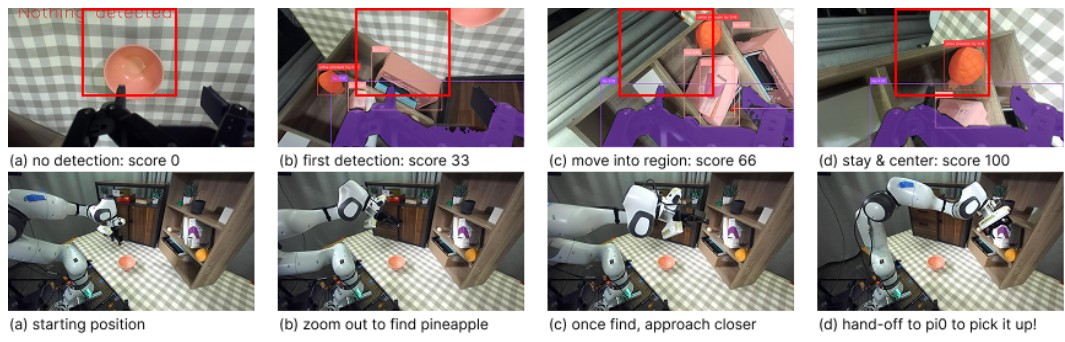

Figure 8: The Search % metric gives points for spotting, approaching, and fixating on the object.

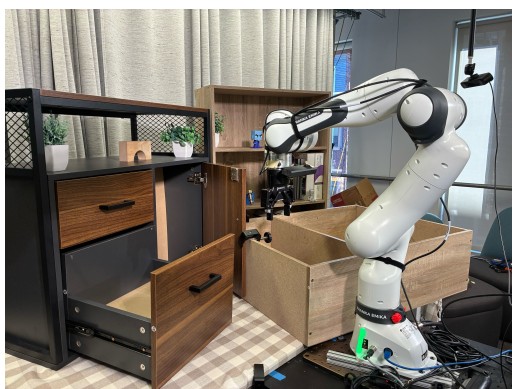

Figure 9: Robot hardware configuration: Franka Panda arm with wrist & side view camera.

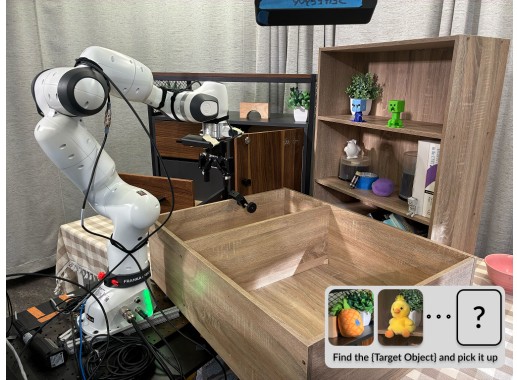

Figure 10: The Complex task has heavy occlusion in the left cabinet, and bookshelf on the floor.

## F.2 Metrics

**Search %**: a 3-point rubric for grading search behavior, see Figure 8 for an example.

1. Policy spots the target object anywhere in the image **[33%]**.

2. Policy moves until target object falls into the target region of viewpoint **[66%]**.

3. Policy fixates on the target object inside the target region **[100%]**.

**Completion**: the grasping success rate of $\pi_0$ (100 timestep limit), after switching from the active perception policy.

**Steps**: mean number of steps for the policy to complete the first two stages of the rubric (finding and approaching). Episodes that fail to reach the first two stages count as a timeout ($T_{max} = 300$ steps).

In Table 5, we also compute normalized metrics to take into account time efficiency, by dividing Search and Completion by the Steps. We then compare all methods relative to the Exhaustive baseline, by dividing the time-normalized Search and Completion metrics by the Exhaustive baselines' time-normalized Search and Completion.

## F.3 Hardware and Scene Setup

We used the DROID robot setup[56], which consists of a 7 DoF Franka Emika Panda Robot Arm, a Robotiq 2F-85 parallel-jaw gripper, a wrist-mounted ZED Mini RGB-D camera and two side-mounted ZED 2 stereo cameras. The DROID set-up enables the usage of the generalist VLA policy $\pi_0$ [57], specifically the FAST-DROID checkpoint.

### F.4 Policy Design

1. **Observation Space**:
   - **Partial observation**: wrist RGB 84×84 image, end effector position of last 6 timesteps, occupancy grid feature.
   - **Privileged observation**: Segmentation mask and bounding box of the target object.
2. **Action Space**: Cartesian and angular velocities of end effector frame,
   $a_t = [v_x, v_y, v_z, \omega_{\text{roll}}, \omega_{\text{pitch}}, \omega_{\text{yaw}}]$. We use action chunks of length 5.

The search policy and privileged critic networks are constructed using an encoder / head architecture. We first detail the input processing steps below, as they are shared for AAWR, AWR and BC network:

The wrist image is first fed into a frozen DINO-V2[58] encoder (ViT-S14) , and the resulting DINO-V2 features are reduced into a $256 \times 16$ dimensional latent using PCA. Next, the occupancy grid feature is constructed by projecting the historical camera rays (inferred through the gripper position) into the XZ dimension of the robot frame, resulting in a 2D occupancy grid where elements are 1 if the camera ray has passed through it in the episode.

The search policy takes in the wrist image features, a history of the last 6 gripper positions, and the occupancy grid feature. The occupancy grid is first processed through a convolutional encoder, and then is concatenated alongside the gripper position history and wrist image features. This latent is then fed into a small MLP to generate the action prediction.

The privileged critic networks also use the same inputs as the search policy, and take in an additional privileged target object segmentation mask. The mask is processed using a small convolutional encoder, and is concatenated with the other inputs.

To handoff from the search policy to $\pi_0$, we implement the following logic. Every 5 timesteps, we query an object detector to see if a target object is detected. If the target object detected in the previous query from 5 timesteps ago and the current query, then we handoff to $\pi_0$. This consecutive mechanism was implemented to prevent premature switching to $\pi_0$, since the object detector is not perfect and sometimes gives false positives. We find that the consecutive criteria rules out false positives and switches correctly when the target object is within view.

### F.5 Reward Design

The reward function incentivizes the robot to locate, approach, and fixate on the target object. Viewpoints that score highly under this reward function feature the object prominently in the top-center region of the wrist image. We chose to incentivize putting the object in this target region because the grippers occupy the bottom half of the wrist image, and find that this particular viewpoint optimizes for grasping success of $\pi_0$.

To define the reward, we first need privileged information about the object location and size in the wrist view image. To obtain object detection and segmentation of the target object, we used the DINO-X [59] API and the GroundedSAM [60] Model for Open-World Object Detection and segmentation. We use a color segmentation mask to break the tie when the object detector detects multiple potential target objects.

During the training phase, we infer the wrist camera images with DINO-X, obtain the bounding box and mask of the target object, and label the reward for offline RL training. See Figure 11 for examples of the object detection and reward pipeline.

The reward function consists of three terms:

(i) **Distance reward**

$$r_{\text{dist}} = 1 - \tanh\left(10 \cdot \frac{D(c, c^*)}{1000}\right), \qquad D(c, c^*) \in [0, 1000] \text{ px},$$

where $c, c^*$ are the centroids of the bounding box, and target region, and $D$ is the L1 distance. This term smoothly saturates to 1 as the bounding box center approaches the image center.

(ii) **Mask–area reward**

$$r_{\text{area}} = \frac{\text{clip}\left(\text{mask\_area}, 1000, 50000\right)}{50000},$$

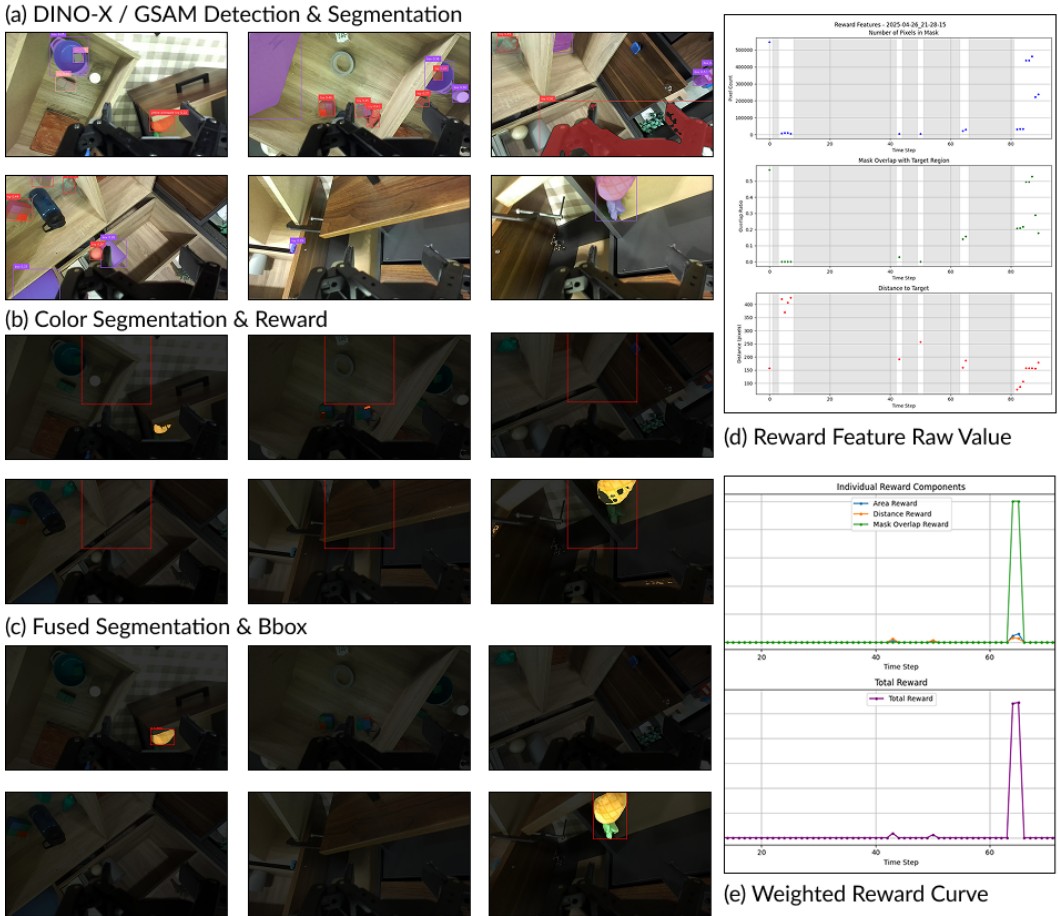

Figure 11: An example reward trajectory in **Complex** scene. Notice the trajectory is given a high peak reward when the detected pineapple overlap on top of the square area.

which is proportional to the mask area subject to lower and upper limits, encouraging the robot to find viewpoints where the object is prominently in view.

(iii) **Mask–overlap reward**

$$r_{\text{overlap}} = \mathbb{1}\big[mask\_overlap > 0.10\big],$$

gives a sparse binary bonus once the object mask intersects with the target region, defined as IoU > 10% with a $128 \times 128$ region centered at $c^*$.

The per-step reward is then composed as

$$\boxed{R_t = 0.5\, r_{\text{dist}} + 0.3\, r_{\text{area}} + 10\, r_{\text{overlap}}},$$

**Demonstrations**  We collect demonstrations using the 5-DoF 3Dconnexion SpaceMouse. During teleoperation, we label trajectories as successful if $\pi_0$ successfully grabs the target object after switching from the teleoperator. The demonstrations are collected with four different teleoperators, with success rates for the demonstrations roughly between 50-70%. We initially collect up to 250 demonstrations per task, but then we curate the dataset, dropping out trajectories with mislabeled object detections, noisy/faulty sensor readings, etc. After filtering, we end up with 152 demonstrations for Bookshelf-P, 109 for Bookshelf-D, 35 for Shelf-Cabinet, and 195 for Complex.

### F.6 Baselines

Please watch the videos on our website (RW-RL Project Page) to better compare the differences among baselines.

1. **AWR**: Advantage-Weighted Regression, no access to privileged observations.

2. **BC**: Filtered Behaviour Cloning, trained on successful trajectories only.

3. **Exhaustive**: Hard-coded baseline that goes over every possible hiding location in a fixed order. This "brute-force" method gets high Search% score but takes much longer to search search.

4. **VLM+$\pi_0$**: This baseline decomposes the task using a VLM for high level task planning and the $\pi_0$ VLA for low level movement as proposed in HiRobot [54]. This approach commonly used in works that solve long-horizon tasks with only foundation models. In practice, we query the Gemini-2.5-Flash [53] model with a task prompt template, which includes a series of searching-related instructions that the low-level VLA can follow. Then, we ask Gemini to choose among them. The prompt template is attached below.

```
You are an expert robot operator using the pi0 policy, a general-purpose robot foundation
    model. It receives a natural language command and executes on the Franka Panda
    robot arm.

Your job is to provide natural-language instructions to help a single-armed robot with a
    parallel-jaw gripper complete a tabletop manipulation task.
---
The overall task is:
find a <|>TARGET_OBJECT_NAME<|> in the scene. The robot has a wrist-mounted camera and
    can perform short sequences of actions, such as opening drawers, scanning
    compartments, and moving its camera viewpoint.

Given the following constraints:
- The <|>TARGET_OBJECT_NAME<|> might be **partially or fully occluded**.
- It could be located **inside drawers**, **behind objects**, or **on shelves**.
---
Your job is to break this task down into smaller instructions that robot can complete
Every few seconds we will ask you to provide a natural language instruction for the robot
    .
We will provide two images: (1) an external view of the robot and (2) a view from a
    camera mounted on the robot's wrist.

Your instruction should refer to relevant objects that you see in the images, and should
    help the robot make progress towards completing the overall task (<|>OVERALL_TASK
    <|>).
To help you, we've prepared a list of instructions for you to choose from:
---
look around for the <|>TARGET_OBJECT_NAME<|>
open the top drawer
open the bottom drawer
look inside the top drawer
look inside the bottom drawer
look behind the toys
look behind the blue block
look behind the green block
look on the top shelf
look on the bottom shelf
move the red block to the side
move the blue block to the side
move the green block to the side
move griper to the right
move griper to the left
move griper to the center
move griper to the front
move griper to the back
move griper to the top
move griper to the bottom
find the <|>TARGET_OBJECT_NAME<|> and pick it up
pick up the <|>TARGET_OBJECT_NAME<|>
---
Here is the external view:
<|>CURRENT_IMAGE<|>
Here is the wrist view:
<|>CURRENT_WRIST_IMAGE<|>
Please provide an instruction for the robot to follow. Write the instruction in all lower
    case with no punctuation. Just provide the instruction; do not provide additional
    explanation.
```

Listing 1: Prompt Template for VLM+$\pi_0$ baseline

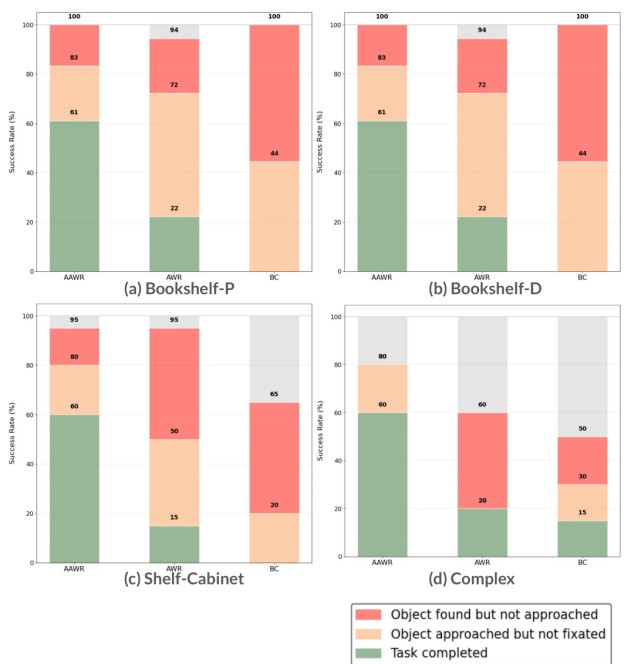

Figure 12: Failure analysis of AAWR, AWR, and BC policies in all 4 tasks. For each policy, we show the number of times each policy completes the first, second and third stage of the Search % rubric. AAWR completes all three stages the most, while AWR and BC fail to consistently approach and fixate on the target object.

Table 4: Metrics for active perception handoff tasks, AAWR consistently outperforms baselines. Bold = best, underline = second best.

| Method | Bookshelf-P | | | Bookshelf-D | | | Shelf-Cabinet | | | Complex | | |
|---|---|---|---|---|---|---|---|---|---|---|---|---|
| | Search % ↑ | $\pi_0$ % ↑ | Steps ↓ | Search % ↑ | $\pi_0$ % ↑ | Steps ↓ | Search % ↑ | $\pi_0$ % ↑ | Steps ↓ | Search % ↑ | $\pi_0$ % ↑ | Steps ↓ |
| **AAWR** | **92.4**±5.0 | **44.4**±16.6 | 36.6±4.7 | 81.3±6.2 | **44.4**±11.7 | 26.9±2.0 | **78.2**±7.0 | 40.0±11.0 | 46.3±4.5 | 54.8±8.5 | 20.0±8.9 | **121.0**±30.1 |
| AWR | 79.6±5.6 | 0.0±0.0 | **34.0**±2.7 | 62.6±6.5 | 16.7±8.8 | 30.2±10.1 | 52.3±6.1 | 10.0±6.7 | **38.0**±13.9 | 13.2±5.0 | 10.0±6.7 | 217.0±29.3 |
| BC | 29.9±13.5 | 20.0±12.6 | 84.0±9.2 | 47.7±4.0 | 16.7±8.8 | **22.5**±2.1 | 28.1±5.5 | 15.0±8.0 | 125.0±29.6 | 46.4±8.5 | 10.0±6.7 | 138.0±30.4 |
| $\pi_0$ | 11.0±11.0 | 16.7±15.2 | 263.3±36.7 | 66.7±21.1 | 33.3±19.2 | 229.7±44.8 | 10.0±10.0 | 10.0±9.5 | 280±20.0 | 29.6±15.3 | 20.0±12.6 | 252.5±31.7 |
| Exhaustive | 64.2±1.8 | 44.0±11.7 | 105.4±9.0 | 96.0±2.7 | 22.2±9.8 | 106.7±8.6 | 52.8±5.0 | 45.0±11.1 | 183.0±15.3 | 78.2±7.8 | 30.0±10.2 | 297.0±30.8 |
| VLM+$\pi_0$ | 31.4±10.2 | 27.8±10.6 | 322.3±31.9 | 33.2±17.1 | 16.7±16.7 | 281.8±18.1 | 28.2±7.3 | 15.0±8.0 | 382±12.6 | 14.8±10.2 | 10.0±9.5 | 374.7±25.3 |

## F.7 Results

As seen in Table 4, AAWR consistently outperforms baselines in all metrics, learning sensible active perception behavior to aid a generalist policy. We report the mean and standard error over 18 trials for all metrics. AAWR always outperforms non privileged AWR and BC, thus validating the usefulness of privileged information and the use of offline RL over supervised learning. The Exhaustive baseline often has high search progress and $\pi_0$% success rate, but is much slower than AAWR, showing that AAWR learns to search efficiently. We find that $\pi_0$ and VLM+$\pi_0$ both search poorly - they tend to take inefficient movements and fail to track the object. In Figure 12, we break down the failures of the active perception policies in each task by recording the number of stages completed in the 3-point rubric. Across all tasks, AAWR completes the task (all three stages) the most. AWR and BC often spot the object, but do not consistently approach and fixate on the object.

In the **Bookshelf** tasks, AAWR first learns to zoom out of the scene to see multiple shelves, then scans from bottom to up, and then approaches the target object once located. The AWR and BC baselines follow a relatively fixed search path that approaches the shelf, but the policies failed to efficiently scan the shelves. Even if they luckily glimpse the target object, they do not fixate on the object, reducing their search score and $\pi_0$ success rate. See Figure 6 for a visualization.

Table 5: Active perception handoff tasks: Metrics normalized by time, and then compared to the exhaustive search. AAWR consistently is ~2−8× better than exhaustive search in such metrics, while other baselines have mixed results.

| Method | Bookshelf-P | | Bookshelf-D | | Shelf-Cabinet | | Complex | |
|---|---|---|---|---|---|---|---|---|
| | Search | Completion | Search | Completion | Search | Completion | Search | Completion |
| **AAWR** | **4.14** | **2.91** | **3.36** | **7.93** | **5.86** | **3.51** | **1.72** | **1.64** |
| AWR | 3.84 | 0.00 | 2.30 | 2.65 | 4.77 | 1.07 | 0.23 | 0.46 |
| BC | 0.58 | 0.57 | 2.36 | 3.56 | 0.78 | 0.49 | 1.28 | 0.72 |
| $\pi_0$ | 0.07 | 0.15 | 0.32 | 0.70 | 0.12 | 0.15 | 0.45 | 0.78 |
| VLM+$\pi_0$ | 0.16 | 0.21 | 0.13 | 0.28 | 0.26 | 0.16 | 0.15 | 0.26 |
| Exhaustive | 1.00 | 1.00 | 1.00 | 1.00 | 1.00 | 1.00 | 1.00 | 1.00 |

In the **Shelf-Cabinet** task, AAWR searches through the right bookshelf, before moving to the left cabinet. Both AWR and BC do not thoroughly search the scene, preventing them from finding objects placed in the left cabinet's drawer.

In **Complex**, AAWR searches the bottom shelf, the right shelf, and then the left cabinet (see Figure 1). See the website for comprehensive success and failure recordings of the policies over all tasks.

### F.8 Dataset ablation on the Complex task.

We ablate the demonstration quality by collecting two demonstration datasets. First, a suboptimal dataset of 195 demonstrations is collected by four different teleoperators. The dataset is quite diverse, suboptimal (success rate of 60%), and potentially hard to learn behaviors from. We found that all policies trained on the initial dataset struggled to reproduce certain behaviors, in particular moving to the left side of the scene, despite the existence of such trajectories in the dataset.

Next, we collected a small 50 trajectory dataset from only one expert teleoperator, with a high success rate of 94%. As seen in Table 6, all approaches improve using the smaller, more optimal optimal dataset. AAWR still outperforms baselines, as it consistently approaches and fixates on the objects, maximizing the success rate of $\pi_0$ after handoff. In contrast, AWR and BC do not approach and fixate target object as well as AAWR, often switching to $\pi_0$ when the target object is barely in view or in an odd location with respect to the gripper.

Table 6: We ablate the demonstration dataset of the Complex task, and find that all approaches benefit from a smaller but more optimal demonstration source. AAWR still outperforms all baselines.

| Method | Complex (Suboptimal Demos) | | | Complex (Expert Demos) | | |
|---|---|---|---|---|---|---|
| | Search % (↑) | $\pi_0$ % (↑) | Steps (↓) | Search % (↑) | $\pi_0$ % (↑) | Steps (↓) |
| AAWR | 54.8 | 20.0 | **121.0** | 73.2 | **50.0** | **43** |
| AWR | 13.2 | 10.0 | 217.0 | 33.2 | 40.0 | 67 |
| BC | 46.4 | 10.0 | 138.0 | 31.5 | 15.0 | 77 |
| $\pi_0$ | 29.6 | 20.0 | 252.5 | 29.6 | 20.0 | 252.5 |
| Exhaustive | **78.2** | **30.0** | 297.0 | **78.2** | 30.0 | 297.0 |
| VLM+$\pi_0$ | 14.8 | 10.0 | 374.7 | 14.8 | 10.0 | 374.7 |

### F.9 Reward Quality Analysis

We further analyze how reward signal quality and annotation noise affect search behaviors. In experiments, all episode begins without the target object in view, and the robot must actively search until it detects the object with visible and confident masks. Ideally, the object is in a clear view see Figure 11, where we give a high terminal reward. Consequently, early reward spikes typically indicate mislabeled frames.

For each dataset, we label the mislabeled episode using a heuristic. An episode is flagged as mislabeled if its reward exceeds a fixed threshold of 5.0 within the first 30% timesteps. We counted the mislabeling rate of success trials, failure trials and total trials of each search datasets below:

Table 7: Per-task policy performance and reward noise.

| Task | Search % (↑) | Completion % (↑) | Mislabel % (Succ./Fail/Total) | Observation |
|---|---|---|---|---|
| Bookshelf-P | $92.4 \pm 5.0$ | $44.4 \pm 16.6$ | 4.6 / 0 / 2.7 | Clean reward |
| Bookshelf-D | $81.3 \pm 6.2$ | $44.4 \pm 11.7$ | 75 / 56.8 / 68.8 | Noisy reward |
| Shelf-Cabinet | $78.2 \pm 7.0$ | $40.0 \pm 11.0$ | 17.1 / 30.8 / 23.8 | Harder scene with longer horizon |
| Complex: Suboptimal | $54.8 \pm 8.5$ | $20.0 \pm 8.9$ | 13.4 / 11.8 / 12.8 | Diverse dirty data. |
| Complex: Expert | $73.2 \pm 12.0$ | $50.0 \pm 15.8$ | 6.4 / 0 / 6.0 | Expert clean data. |

As the table shows, higher quality data (e.g. clean, sparse reward with less detection error) correspond to better search and completion rates. Despite noisy rewards in Bookshelf-D, AAWR is still able to learn good search behavior.

## G  Blind Pick Details

**Task Definition**  In this real world experiment, the Koch robot must pick up a small rectangular candy (1 cm × 1 cm × 2 cm). It operates blindly since it is given only the initial object position and its own joint positions during the episode. Interactive perception is required to solve the task, since the robot must sense when the object is gripped using its proprioception and then proceed to lift it up.

- **Observation**: Initial object position, and current robot joint positions.
    - **Partial observation**: joint positions and initial Cartesian position of the target.
    - **Privileged observation**: real time Cartesian position of the target at each timestep.
- **Action Space**: Cartesian position commands relative to robot base and gripper joint control.

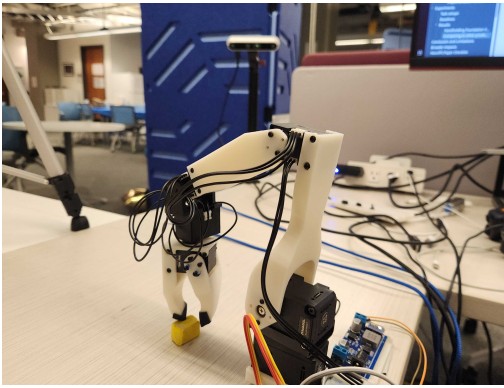

Figure 13: Hardware configuration: Koch robot with RGB-D camera in the back.

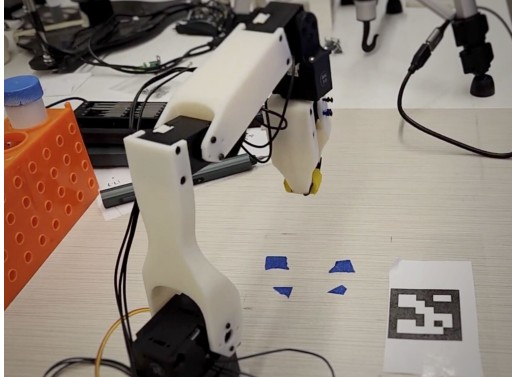

Figure 14: Koch robot picking up the target object.

**Hardware and Scene Setup**  We utilized a Kochv1.1 robot[61], an open-source, low-cost, 5 DoF robotic arm. The robot arm is operated via a Cartesian position controller respect to the robot base frame. The forward and inverse kinematic computations is computed in a MuJoCo simulation model synchronized with real robot's joint positions in real-time.

To get the privileged cartesian position of the object, we set up a RealSense D455i RGBD camera pointed towards the robot workspace. We calibrate the D455i using an ArUco marker, and then use color segmentation to filter the point cloud to estimate the 3d position of the object on the table. The target object is randomly placed within a 10cm square region in front of the robot at the start of each trial.

**Data Collection** Data for the Koch robot experiment was collected by executing approximately 100 demonstration episodes, in total containing around 3000 transitions. The demonstrations were gathered from a noisy hand-coded script, resulting in a success rate of approximately 20%.

**Reward Design**

1. **Distance penalty** The distance penalty is the reward term where we introduce privileged information: the real-time position of the target object. Th reward term is computed by:

$$r_t = - \left\| x_t - x^* \right\|,$$

where $x_t$ is the current Euclidean position of the target object computed via color segmentation and the depth camera, $x$ is the current Euclidean position of the robot's end effector.

2. **Grasp reward** Using robot proprioception, we can determine if the gripper has a firm grasp on an object. More specifically, if the gripper receives a closing command, but the actuator cannot rotate the gripper to the commanded position, a firm grasp is detected by proprioception. Therefore, we have:

$$r_{grasp} = k_{grasp} \mathbb{1}_{\{\text{grasped}_t = \text{True}\}}$$

3. **Success reward** A larger reward when the robot fully accomplishes the task: picking up the target object and lifts it 7 cm above the robot's base. In particular, the reward is given by:

$$r_{success} = k_{grasp} \mathbb{1}_{\{z_{ee} - z_{base} > 0.07 \wedge \text{grasped}_t = True\}},$$

where $z_{ee}$ is the z-axis position of the end effector, $z_{base}$ is the z-axis position of the robot base.

**Baselines**

1. **BC**: Behavior Cloning using offline successful demonstrations only.
2. **AWR**: Advantage Weighted Regression trained both offline and online.
3. **AAWR**: Asymmetric Advantage Weighted Regression leveraging privileged information during offline and online training phases.

**Metrics** Performance is evaluated across 40 trials per method with the following metrics:

1. **Grasp %**: Percentage of trials in which the robot successfully grasped the object.
2. **Pick %**: Percentage of trials where the robot successfully grasped and lifted the object.

**Online Training and Evaluation** All methods underwent an initial 20,000-step offline pretraining phase followed by online fine-tuning using 1,200 transitions ( 40-50 episodes), each lasting approximately 20 minutes.

During evaluation, for each trial, a policy has 30 timesteps to accomplish the task. Evaluation results confirmed AAWR's superior performance in grasping and picking success rates and demonstrated notably effective retrying behaviors following failed initial attempts.

## H    Simulated Experimental Details

### H.1    Camouflage Pick

The Camouflage Pick experiment requires a simulated Koch robot to pick up a tiny, hard-to-see marble initialized in a 10×10 cm region (see Figure 15).

- **Observation Space**:
    - **Partial observation**: 3rd person 84×84 image of the robot and marble.
    - **Privileged observation**: robot and marble positions using simulator state

- **Action Space**: Cartesian velocity of end effector frame and gripper position
- **Reward Function**: Sparse reward that gives $+1$ if the marble is in gripper and altitude is over 7cm.
- **Demonstrations**: We collect 100 demos using a hand-coded script. The script is not perfect and gets around 30% success rate.
- Offline / Online budget: 20K offline, 80K online

|  (a) AAWR | (b) AWR | (c) BC |
| :---: | :---: | :---: |

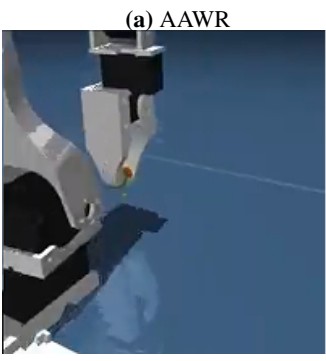 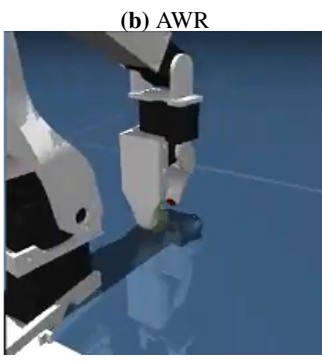 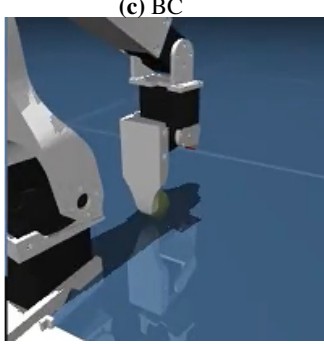

Figure 15: **Camouflage Pick (third-person camera only).** (a) **AAWR**: succeeds with search then grasp. (b) **AWR**: fails due to mis-grasps. (c) **BC**: fails due to overfit on grasping without search

We compare against symmetric AWR, the non-privileged version of AAWR, and BC. We train BC for 20,000 offline steps, periodically checkpointing and evaluating it, and report the highest performing checkpoint. We pretrain AWR and AAWR for 20,000 offline steps. Then, we do online finetuning for 80,000 environment steps. While training, we periodically evaluate the policies by recording their average success over 100 trials. The success metric is the sparse reward function.

We train all models with a batch size of 256, learning rate of 0.0001, and the Adam optimizer. For online finetuning following [47], we use an update-to-date ratio of 1 , performing gradient updates after every episode. For AWR and AAWR, we use an advantage temperature of 10.

We instantiate separate networks for the for the policy and value/critic networks. We use the same encoder / head recipe for all models, following [47]. We use a CNN to process the RGB image into a 50-dimensional latent, and a MLP to process the privileged information into a 50-dimensional latent. The latents are then fed into a MLP to get the output.

As seen in Figure 5, AAWR outperforms baselines. AWR and BC frequently completely miss the marble, while AAWR displays more accurate picking behavior. Even after picking, the small marble frequently slips out of the grasp, making the success rate for all of the policies rather low. See the website for videos.

## H.2 Fully Observed Pick

The Fully Observed Pick experiment requires a simulated xArm6 robot to pick up a block. To make the scene as fully observable as possible, we make the xArm6 robot invisible except for its grippers, making occlusion of the object by the robot impossible (see Figure 16). The object is randomly initialized in a $25 \times 25$cm region in front of the arm.

- **Observation Space**:
    - **Partial observation**: 3rd person $84{\times}84$ image of the robot and block.
    - **Privileged observation**: robot and object positions using simulator state
- **Action Space**: Cartesian velocity of end effector frame and gripper position
- **Reward Function**: Sparse reward that gives $+1$ if the block is in gripper and altitude is over 10cm.
- **Demonstrations**: 100 demos using a hand-coded script. The script is not perfect and gets around 30% success rate.
- Offline / Online budget: 20K offline, 20K online

We use the same baselines, hyperparameters, network, and evaluation configuration as the Camouflage Pick experiment. The only change is the offline / online budget - 20K steps offline, and 20K steps online.

(a)  (b)  (c)  (d)

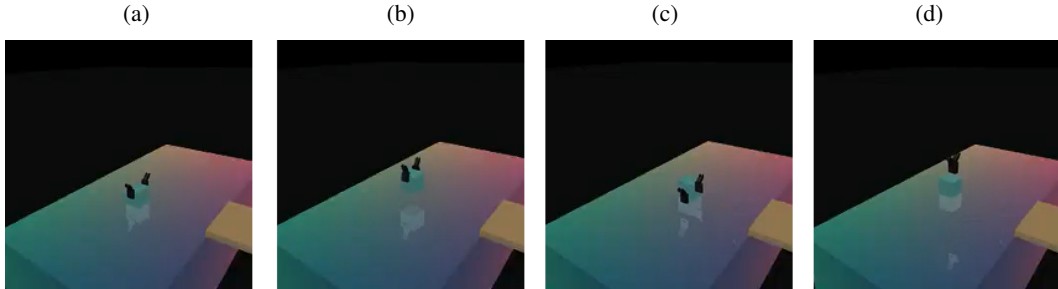

Figure 16: Visually ambiguous pick (fully observed). (a–b) **AAWR**: grasp centers then lifts (success). (c–d) **AWR**: hovers off-center, closes mid-air (fail).

Results are in Figure 5. AAWR effectively solves the task with near perfect success rate, learning to accurately localize and grasp the object. AWR shows two failure modes. First, it struggles with positioning the gripper over the block. It often places the gripper in front of the block, which looks like a reasonable grasp from the camera angle, but is in reality quite off from the block (see Figure 16). Next when it is able to grasp the block, it does not lift up the block. BC displays similar failure modes. See the website for videos of the policies.

### H.3  Active Perception Koch

In this task, we equip the simulated Koch robot with a wrist camera with a small field of view, and task it to pick up a cube randomly initialized in a $10 \times 20$ cm region in front of it. (see Figure 17). As the wrist camera has a narrow field of view and becomes self-occluded during grasp closure, the robot must first scan to rediscover the cube before executing the pick.

- **Observation Space**:
    - **Partial observation**: Frame-stacked (past 3) grayscale wrist camera images of size $84 \times 84$.
    - **Privileged observation**: object positions using simulator state
- **Action Space**: Cartesian velocity of end effector frame and gripper position
- **Reward Function**: Sparse reward that gives $+1$ if the object is in gripper and altitude is over 7cm.
- **Demonstrations**: We collect 100 demos using a hand-coded script. The scripted behavior uses state information to command the robot to go over the block and pick it up. The script is not perfect and gets around 30% success rate.
- Offline / Online budget: 100K offline, 900K online

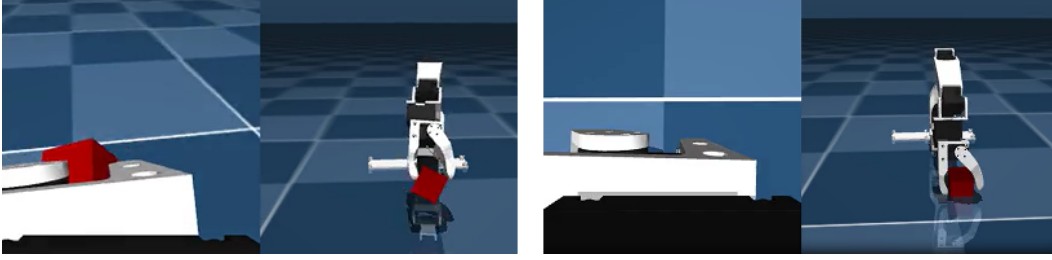

Figure 17: **Wrist camera with limited field of view.** When the gripper approaches and closes on the cube, the wrist camera becomes self-occluded, The side view is shown for visualization/evaluation only and is not provided to the policy.

We compare against other approaches that use privileged information. The first is **Distillation** [39, 62], which features a two-stage training process. In the first stage, teacher acquisition, a privileged teacher

policy is trained on the collected successful demonstrations. In the second phase, distillation, the teacher is distilled into a student policy. Following [62], the distillation phase is performed over online rollouts from the student policy. In our setup, after the first stage, the teacher policy is able to get near perfect success rate on the task, although note that it is using privileged information to do so.

The second baseline is a variational information bottleneck approach **VIB** [52] that trades off the RL return with a KL penalty for accessing privileged information. Concretely, this penalty is implemented by defining a privileged latent that comes from the posterior $z \sim p(\cdot \mid o^+)$, and constraining the posterior to the unprivileged gaussian prior $\mathcal{N}(0, I)$ via KL divergence. During training, the policy $\pi(a \mid o, z)$ uses the latent from the privileged posterior, and during evaluation uses a latent sampled from the unprivileged prior. The RL agent should learn to minimally use privileged information, since usage will negatively impact its overall return. We conduct sweeps over different weights of the KL term $\beta = 0.01, 0.1, 0.5, 1, 10$, report the performance of the best performing weight ($\beta = 0.5$).

All baselines are implemented in the same codebase, using the same encoder / head architecture configuration as AAWR. We conduct sanity checks to make sure the baselines work, such as making sure the privileged teacher and VIB policy with the privileged latent get high success rates.

| (a) AAWR | (b) Distillation | (c) VIB |

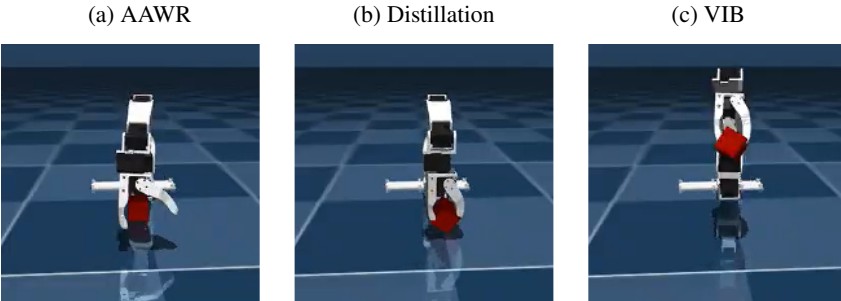

Figure 18: **Resulting behaviors on the Koch task.** (a) **AAWR** actively scans the workspace, recenters the cube in view, grasps, and lifts with near 100% success at evaluation. (b) **Distillation** learns a suboptimal "go-to-center" strategy and often closes off-target due to the absence of scanning in the teacher. (c) **VIB** degrades at evaluation without privileged information; using only a prior latent leads to drift and low success.

As seen in Figure 18, only AAWR learns to do active perception by scanning the workspace, getting near 100% success rate during evaluation time. The distilled student learns a suboptimal behavior of just approaching the center of the workspace, because its privileged teacher never displays the scanning behavior. VIB does poorly during evaluation with no access to privileged information, even though it was trained to minimally use privileged information.

Additional video examples are available on our project page at `https://penn-pal-lab.github.io/aawr/`.

