# OpenReview forum: "Real-World Reinforcement Learning of Active Perception Behaviors"
_NeurIPS.cc/2025/Conference — NeurIPS 2025 poster_

### Official Review · Reviewer_3ct1 · 2025-07-01

**Clarity:** 2
**Significance:** 2
**Originality:** 2
**Rating:** 5
**Confidence:** 2

**Summary:**

This paper proposes an extended version of AWR, termed Asymmetric AWR (AAWR), for policy improvement in POMDP settings with access to privileged information during training. The goal is to leverage this privileged information to promote active perception behaviors. The main novelty on the algorithmic side lies in the asymmetric structure: the critic has access to privileged information (e.g., additional observations such as position, masks, or bounding boxes), while the policy is trained using only standard observations. The method is evaluated on both simulated robotics tasks and real-world robot learning tasks, with comparisons to vanilla BC, the original AWR, active perception baselines, and VLA-based models.

Overall, I find the proposed method to be simple and practical, and the experiments are well-executed with a set of realistic, real-world benchmarks.

**Questions:**

I have listed my questions (Q1–Q5) in the weakness section above, as they are closely related to the identified weaknesses and grouping them together improves coherence. In the rebuttal, it would be helpful to see clear answers to Q1, empirical validation for Q3, a discussion addressing Q2, and concrete plans for how the authors intend to revise the paper to address Q4.

**Ethical Concerns:**

["NO or VERY MINOR ethics concerns only"]

**Final Justification:**

All my concerns have been addressed in the rebuttal, and I also checked the rebuttal to other reviewers - all look sensible to me.

**Limitations:**

Yes

**Quality:**

2

**Strengths And Weaknesses:**

**Strengths**

(+) [About novelty and significance] The problem setting itself is important. While there has been prior work on learning active perception, this paper applies AAWR to address it, and I feel that the algorithmic novelty is reasonable within this context.

(+) [Evaluation] I think the paper provides a solid set of experiments, covering both simulated and real-world scenarios, and overall, the evaluation is convincing. That said, the inclusion of additional relevant baselines could further strengthen the empirical results.

(+) [Clarity] In general, the writing is clear and easy to follow, and most of the necessary details are provided. I also checked the appendix, which includes the full derivations of the objective functions as well as comprehensive experimental details.

**Weaknesses and questions**

**W1** [Model] In general, I don't find specific technical problems in the proposed AAWR, however, I do have questions on the setting:

*Q1*: What is the fundamental difference between the proposed method and existing approaches that learn active perception by introducing an additional motor policy? In those cases, privileged information is also often used to train the critic. One example is Shang et al., 2023.

*Q2*: Is it possible to allow the agent to actively select the privileged information to use also during training time, rather than relying on a pre-specified set during policy improvement?

**W2** [Evaluation]

*Q3*: Similarly to Q1,I think it would be sensible to also compare with Shang et al., 2023 to empirically demonstrate the differences. If this is not feasible within the current timeframe or it is not doable, providing a justification in the rebuttal would be sufficient.

**W3** [Presentation] I think the presentation and clarity could be further improved:

*Q4*: I feel it would be sensible to present the AAWR component with more details. At a minimum, the beginning of Section 3 should explicitly define the problem setting, clearly distinguishing what constitutes privileged information, what is considered sensory input, what the final objective is, and conceptually, how the side information is used during training.

*Q5*: Minor: Line 147, V(s,a) -> V(s,z); Line 188: missing citation.


Shang, Jinghuan, and Michael S. Ryoo. "Active vision reinforcement learning under limited visual observability." Advances in Neural Information Processing Systems 36 (2023): 10316-10338.

---

> ### Author Rebuttal · Authors · 2025-07-31
>
> Thank you for your comments!
>
> > Q1: What is the fundamental difference between the proposed method and existing approaches that learn active perception by introducing an additional motor policy?
> > Q3: …it would be sensible to also compare with Shang et al., 2023 to empirically demonstrate the differences
>
> **Shang et al. requires a factorized action space, while AAWR does not.** The fundamental difference between AAWR and such prior work like Shang et al. [1] is that Shang et al. only works on POMDPs with a “factorized” action space (explained below), whereas AAWR is derived with general POMDPs in mind, including those without factorized action spaces.
>
> Specifically, Shang et al. requires that the action space can be factorized into “sensing” actions that move the sensor without affecting the scene, and “motor” actions that move the robot without affecting the sensor’s viewpoint. In their simulated robotic tasks, they assume access to a 3rd-person camera that can freely move around the scene to observe the robot. This assumption is mentioned in their limitation section, where they mention future work is needed to address tasks where there is a dependence between sensing and motor actions.
>
> **AAWR tasks cannot be factorized.** The tasks evaluated in AAWR do not meet Shang et al.’s requirements. In our active perception tasks, the only moving camera is a wrist camera that is attached to the same robot arm that must manipulate the scene. This is very common in modern manipulation setups including diffusion policies, VLA models such as pi-zero, and more. In these settings, any action will simultaneously change the viewpoint and affect the scene. To take another example from our paper, in our Blind Pick task, the robot needs to perform *interactive* perception by grasping a block to detect it before picking it up. There is no clear distinction between sensor and motor actions, since the robot needs to physically affect the scene by grasping the block to detect it. Since AAWR does not assume a factorized action space, it can solve these tasks, while Shang et al. cannot.
>
>
> **Shang et al. is not ready for real world RL.** Aside from assumptions in task formulation, extensive algorithmic modifications would be needed to make Shang et al. work in the real world. Shang et al. was trained only in simulation with online RL. In contrast, AAWR learns strictly in the real world from offline and online data, with a sample budget several orders of magnitude below the sample budgets of Shang et al.
>
> In summary, we do not think it is feasible to apply Shang et al. to our tasks since our tasks’ action spaces cannot be factorized, which is a key assumption of Shang et al. Additional and substantial algorithmic modifications would be needed to make it more sample efficient for real world learning. Extending Shang et al. to real world tasks, potentially using ideas introduced in AAWR to make it efficient, would be an interesting future direction, but is not feasible within our rebuttal timeline.
>
> > In those cases, privileged information is also often used to train the critic. One example is Shang et al., 2023.
>
> After reading Shang et al. 2023, we could not find any equation or text mentioning a privileged critic. It is possible we missed something, could the reviewer point to where it mentions the usage of a privileged critic?
>
> > Q2: Is it possible to allow the agent to actively select the privileged information to use also during training time, rather than relying on a pre-specified set during policy improvement?
>
> We agree that this would be a cool direction to pursue! In fact, this is close to our own motivations for pursuing this work, and so we’ve already  mentioned it as a potential future direction in the conclusion in line 349. One major challenge we anticipate is learning efficiency. Learning to choose sensors through RL might be  too sample-inefficient for real-world settings. We believe alternative strategies from the active feature acquisition [2] and co-design [3]  literature could be promising ways to perform sensor selection efficiently.
>
> > Q4: I feel it would be sensible to present the AAWR component with more details.
>
> Thank you for the feedback on the presentation, and here’s our plan for improving clarity.
>
> > …the beginning of Section 3 should explicitly define the problem setting, clearly distinguishing what constitutes privileged information, what is considered sensory input,
>
> - Line 104: The paragraph before motivated modelling active perception tasks as POMDPs. At the start of this paragraph, we will explicitly define the problem setting as “finding a policy that maximizes return in such POMDPs”.
> - Next, we will clearly define what is the sensory input and privileged information. The sensory input is the observation space of the POMDP. The privileged information is either the hidden state of the POMDP or additional observations that are informative of the state.
>
> > what the final objective is, and conceptually, how the side information is used during training.
>
> - The final objective is equation 3, and we will explicitly write that it is the final objective of our method. To make it clear where the privileged information is used, we will highlight the privileged input $s$ in blue in equation 3,  and in lines 147-148.
> - In the next version of the manuscript, we will add a figure visualizing the computation graph of equation 3. It will show two components, a policy and a privileged critic. The policy will just take in the observation, while the privileged critic will take in the observation and privileged information.
>
> > Q5: Minor: Line 147, V(s,a) -> V(s,z); Line 188: missing citation.
>
> Thank you, we will update it in the next version of the manuscript.
>
> [1] Shang, Jinghuan, and Michael S. Ryoo. "Active vision reinforcement learning under limited visual observability." Advances in Neural Information Processing Systems 36 (2023): 10316-10338.
>
> [2] Rahbar, Arman, Linus Aronsson, and Morteza Haghir Chehreghani. "A Survey on Active Feature Acquisition Strategies." arXiv preprint arXiv:2502.11067 (2025).
>
> [3] Antonova, Rika, and Jeannette Bohg. "Learning to be multimodal: co-evolving sensory modalities and sensor properties." Conference on Robot Learning. PMLR, 2022.

---

> > ### Comment · Reviewer_3ct1 · 2025-08-01
> >
> > Thank you for your detailed response. All my concerns have been addressed. Regarding Shang et al., I now understand the differences more clearly and acknowledge my earlier misunderstanding about the use of privileged information in their work. As for Q2, I’m excited about this future direction. Good job, and I have updated my score accordingly

---

### Official Review · Reviewer_u7RD · 2025-07-02

**Clarity:** 4
**Significance:** 3
**Originality:** 3
**Rating:** 5
**Confidence:** 2

**Summary:**

This paper presents a method called Asymetric Advantage Weighted Regression that allows for training of active perception policies under partial observability. It uses information provided by sensors during training to efficiently train value functions that are more accurate for providing advantage estimates. The method extends Advantage Weighted Regression to this type of setting, in which the deployed policy has asymmetric perception capabilities. The paper tests this method on a variety of simulated and real world tasks and find significant improvements over baseline techniques.

**Questions:**

1. Do the authors have a sense of how "clean" the privileged information has to be for the method to be useful?
2. How much does the type of privileged information impact the learned policies (for a given task)?
3. Do the failure modes of the method likely stem more from the demonstration noise or privileged information noise?

**Ethical Concerns:**

["NO or VERY MINOR ethics concerns only"]

**Final Justification:**

I feel that this paper should be accepted, as it has a high standard of quality of work and my concerns have been addressed by the rebuttals.

**Limitations:**

Yes

**Quality:**

4

**Strengths And Weaknesses:**

Strengths:
1. The paper is very well-written and clear, and presents a clear motivation for their method in the context of integrating active perception with techniques using foundation VLA models.
2. The real-world experiments are very helpful in illustrating the advantage of their method.
3. Although I am not very well-versed in this area (and therefore this comparison may not be accurate), the method itself seems efficient in terms of clock time and number of demonstrations needed for training.

Weaknesses:
1. I'll add the caveat that this particular area of active perception is not my strongest expertise, it seems that the novelty of the work is rather incremental, as the bedrock of the method is AWR, an existing method.
2. There is a lack of comparison with other approaches that use privileged information as mentioned in related work.

---

> ### Author Rebuttal · Authors · 2025-07-31
>
> Thank you for the feedback!
> > novelty may be incremental relative to AWR?
>
> As we argued around line 160 in the paper, Advantage Weighted Regression (AWR) was originally formulated for fully observed settings. One contribution of ours is showing that its naive application to POMDPs would result in suboptimal learning, see lines starting from 160 for an intuitive explanation and full theorem in Appendix D. We then show both theoretically and empirically why AAWR is the right instantiation of AWR for partially observed tasks
>
> AAWR is a mechanistically simple modification of AWR, but it is precisely the simplicity of implementation that makes the AAWR recipe easy to evaluate on many diverse tasks, including with real-world robots, and alongside state-of-the-art generalist robot policies.
>
> > There is a lack of comparison with other approaches that use privileged information as mentioned in related work
>
> Our task setup involves learning directly on a real world robot platform, with no access to a simulator. Due to the slow data collection time of real robots, the sample budget is very small (e.g. two hundred offline trials, and 50 online trials). Furthermore, the privileged observations are estimated using noisy, real world sensors.
>
> True, we did indeed mention other privileged RL approaches in related work, but as we have argued there (Lines 82-95), these approaches are designed for sim-to-real RL, and (1) sim-to-real RL is not easily applicable to active perception due to the difficulty of effectively simulating most sensors, and (2) the privileged RL approaches designed for sim-to-real do not easily transfer to our settings, because they assume access to millions to billions of episodes of training data, and exploit low-dimensional ground truth simulator state as privileged information. As such, these methods are not empirically comparable to ours.
>
> > Do the authors have a sense of how "clean" the privileged information has to be?
>
> We have some empirical sense of the robustness of our method to noise in the privileged information. In our real world experiments, we show that AAWR is able to learn from privileged observations, which have some irreducible noise due to imperfect sensing and environmental constraints.
>
> For example, in the active vision pi0 experiments, we used off-the-shelf object detectors [1,2] to get privileged segmentation masks of the target objects (see figure 10 in appendix). These detectors would sometimes misclassify objects (e.g. classifying distractor objects with similar appearances as the target class), with a non-trivial misclassification rate of ~22%. We will include more discussion and details of the privileged sensor noise in the experimental detail sections (appendix F and G).
>
> Next, in the Blind Pick experiment, we used a depth sensor (Realsense D435i) to get 3D position estimates of the object. On average, the estimates were off by 0.5-1cm, which is  non-trivial considering the small size of the object (around 1.5cm) and the workspace (around 10cm). There are also cases when the 3D position estimates are invalid, such as when the robot gripper occludes the object.
>
> It may also be interesting to try to theoretically characterize the sensitivity / robustness of AAWR to noise in the privileged observations. We will look into this in future work.
>
> > …does the type of privileged information impact policies?
>
> Due to our limited resources on running real world RL experiments, we did not test different types of privileged observations for a given task. But we can give a few guidelines for choosing privileged information.
>
> First, for the privileged critic, the privileged information should aid in predicting returns. This implication is drawn from appendix C and D, where we show the advantage is determined by both the agent state and environment state. In our active perception tasks, the location of the hidden object is an important privileged feature since the return is dependent on the object location.
>
> Next, one should consider how easy it is to learn from the privileged sensor. For example, high dimensional sensors like cameras and depth sensors would require higher capacity encoders, which in turn demand more data samples to learn good representations.
>
> > Do the failure modes … stem more from the demonstration noise or privileged information noise?
>
> We observe that in the offline case, the policy is more dependent on the particularities of the demonstrated behaviors, since it cannot improve from on-policy trajectories.
>
> In the online RL phase, demonstration noise is no longer a problem since the agent is learning from its own on-policy trajectories. Here, we believe privileged information noise may impact the performance more, since noisy privileged information would result in less accurate advantage estimates, which would affect the policy gradient.

---

> > ### Comment · Reviewer_u7RD · 2025-08-05
> >
> > Thank you to the authors for this comprehensive response. My concerns have been adequately addressed and I will update scores accordingly.

---

> ### Comment · Area_Chair_ShiF · 2025-08-05
> **Please respond to the authors' response**
>
> Dear reviewer u7RD,
>
> Thanks for your reviewing efforts so far. Please respond to the authors' response.
>
> Thanks,
> Your AC

---

### Official Review · Reviewer_CA6i · 2025-07-03

**Clarity:** 2
**Significance:** 3
**Originality:** 3
**Rating:** 4
**Confidence:** 3

**Summary:**

The paper proposes Asymmetric Advantage Weighted Regression (AAWR), a reinforcement learning method for training robots to perform active perception behaviors in real-world environments with partial observability. AAWR leverages "privileged" extra sensor information available only during training to train value functions that provide better supervision signals for the policy. The approach combines a small number of suboptimal demonstrations with a coarse policy initialization and uses offline-to-online RL to efficiently acquire information-gathering behaviors. The method is evaluated on eight manipulation tasks involving different robots under varying sensory setups. The results show that AAWR learns active perception policies and outperforms baselines.

**Questions:**

Practically, if or why AAWR would be better than some search-based heuristics in active perception in your tasks performed? Or are there other tasks that cannot be simply done by heuristics search but well solved by AAWR?

**Ethical Concerns:**

["NO or VERY MINOR ethics concerns only"]

**Final Justification:**

I've read the author response. While I still do not agree with some of the comments, I would like to sync with fellow reviewers and remain on the accept side.

**Limitations:**

Partially, but limitations are not sufficiently addressed.

**Paper Formatting Concerns:**

No.

**Quality:**

3

**Strengths And Weaknesses:**

Strengths:

+ The paper introduces privileged information at training time to enable robots to efficiently learn active perception behaviors. This is a meaningful advancement, as it addresses the sample inefficiency and partial observability issues.
+ The authors provide a solid theoretical justification for their approach, deriving the AAWR objective for POMDPs and clearly explaining why asymmetric advantage estimation is crucial in these settings.
+ Empirical results are strong: Across a variety of tasks and robot platforms, AAWR demonstrates good performance compared to both imitation learning and standard RL baselines.

Weaknesses:

- The validity and generalizability of the approach are somewhat unclear, as the experimental tasks are relatively simple or "toy" problems. In many of these settings, a hard-coded exploratory policy or a straightforward exhaustive search could plausibly solve the task. While such a policy might not be efficient at first sight, it might also be justified considering the cost to train and collect data for AAWR.
- The real-world benefit of sophisticated active perception may be less apparent in scenarios where domain knowledge or engineered exploration strategies could also succeed. The comparison to such baselines (e.g., exhaustive search or heuristic exploration) is mentioned, but the gap between learned and non-learned approaches may not be as meaningful in these simplified settings.

---

> ### Author Rebuttal · Authors · 2025-07-31
>
> We thank the reviewer for their helpful feedback, and discuss their concerns below. The reviewer asked “why AAWR would be better than **search-based heuristics** in …your tasks?”. They raise the point that non-learning-based approaches like a “hard-coded exploratory policy or a straightforward exhaustive search”, could plausibly do well, without requiring data collection and training costs.
>
> Indeed, active and interactive perception tasks are search problems, and all search problems do have brute force “solutions” that will eventually succeed. The measure of good active and interactive perception strategies is how *efficiently* they can search. This is reflected in our performance metrics, where we measure not just if the policy succeeds at finding the object, but also the number of steps it takes.
>
> AAWR is a learning based approach for acquiring active perception policies, and we seek to learn policies rather than engineer them, for the following compelling reasons.
>
> 1. Unlike a fixed, engineered policy, AAWR, like any other RL algorithm, can improve over time from its own experience towards an optimal policy. In contrast, a heuristic strategy may solve the task, but is likely inefficient. For example, we find that our engineered exhaustive search baseline, and a VLM-powered heuristic search baseline, both take over 3 times longer than AAWR to find the objects.
>
> 2. Engineered solutions often require more expansive sensor setups than what we afford to AAWR in our experimental setups. For example, [1] uses depth sensors and [2] uses calibrated cameras. In contrast, AAWR only uses a single uncalibrated camera in the $\pi_0$ active perception task, and only proprioception encoders in the Blind Pick task.
>
> 3. Engineered heuristic solutions tend to be very specific for each task and sensor setup, and may need to be completely reimplemented if anything changes. AAWR policies are acquired using the same recipe across different robots, sensors, and tasks, through collecting data and optimizing the AAWR objective.
>
> Next, the reviewer is concerned about the scope of our experiments, in that “the experimental tasks are relatively simple or **toy problems**”. Our tasks in sim and in real are relatively hard compared to recent prior work on learning active perception policies [3,4,5], and are thus meaningfully advancing the state of the art. However, as mentioned in the paper, the field as a whole has struggled to produce useful active perception behaviors for robotic manipulation, and this is reflected in the complexity of tasks that are commonly investigated.
>
> AAWR offers an improved technique and meaningfully advances the complexity of active perception behaviors that can be learned for real robotic setups, even showing the capacity to improve the behavior of a state-of-the-art generalist robot policy. Even so, we agree that there is much more room for growth, and hope that our paper will be an important stepping stone in this direction!
>
> We will aim to add this context into the paper, thank you again for the thoughtful queries.
>
> [1] Krainin, Michael, Brian Curless, and Dieter Fox. "Autonomous generation of complete 3D object models using next best view manipulation planning." 2011 IEEE international conference on robotics and automation. IEEE, 2011.
>
> [2] Liu, Yang, Pansiyu Sun, and Akio Namiki. "Target tracking of moving and rotating object by high-speed monocular active vision." IEEE Sensors Journal 20.12 (2020): 6727-6744.
>
> [3] Grimes, Matthew Koichi, et al. "Learning to look by self-prediction." Transactions on Machine Learning Research (2023).
>
> [4] Cheng, Ricson, Arpit Agarwal, and Katerina Fragkiadaki. "Reinforcement learning of active vision for manipulating objects under occlusions." Conference on robot learning. PMLR, 2018.
>
> [5] Dass, Shivin, et al. "Learning to Look: Seeking Information for Decision Making via Policy Factorization."Conference on robot learning. PMLR, 2024.

---

> > ### Comment · Reviewer_CA6i · 2025-08-05
> >
> > Thanks for the clarification. I do not buy some of the arguments that search-based methods have this or that limitations and data-driven is simpler better. Data-driven relies on well-collected data right? In your specific example of opening drawers / cabinets to search problem, data collection seems just much clumsier than a naive search-based method. For harder, more complex problems, that argument might apply.

---

> > > ### Author Response · Authors · 2025-08-06
> > > **Continued discussion on data-driven and heuristic approaches**
> > >
> > > Thank you again for your continued engagement.  To restate and clarify our response above, we do not mean to claim that data-driven approaches are always simpler or better for any one task. Rather, our arguments for methods like ours are as follows.
> > >
> > > First, data-driven policy synthesis offers the promise of a unified solution scalable to many tasks, while manually engineered solutions would rely on task-specific heuristics. This has proven true for general robotic control in recent years, and for active perception, we believe our method advances the state of the art here meaningfully. This allows us to take on more complex tasks than in many prior learning-based active perception efforts [1,2,3], with the ability to synthesize either (a) unified control policies that perform both information-gathering and task-solving actions, or (b) specialized active perception policies that improve the performance of a pre-trained generalist manipulation policy.
> > >
> > > Additionally, we do agree that the relative gains of a data-driven approach are likely to be still higher as the task complexity increases further. As we stated in our first response too, we are indeed still operating in the kinds of relatively simple tasks where one could plausibly engineer a heuristic task-specific search behavior. As task complexity increases, engineering simple heuristics quickly becomes infeasible, such as tasks with conjoined search and task execution structures (e.g. searching for target objects in clutter [4]). In fact, as we stated above, it is already the case that for many of our tasks, previously proposed heuristics for search [5,6] would require more expansive sensor setups than the ones we investigate.
> > >
> > > Thank you for bringing these points up, we will discuss them in the paper.
> > >
> > > [1] Grimes, Matthew Koichi, Joseph Varughese Modayil, Piotr W. Mirowski, Dushyant Rao, and Raia Hadsell. "Learning to look by self-prediction." Transactions on Machine Learning Research (2023).
> > >
> > > [2] Cheng, Ricson, Arpit Agarwal, and Katerina Fragkiadaki. "Reinforcement learning of active vision for manipulating objects under occlusions." In Conference on robot learning, pp. 422-431. PMLR, 2018.
> > >
> > > [3] Dass, Shivin, Jiaheng Hu, Ben Abbatematteo, Peter Stone, and Roberto Martín-Martín. "Learning to Look: Seeking Information for Decision Making via Policy Factorization." In Conference on robot learning. PMLR, 2024.
> > >
> > > [4] Danielczuk, Michael, Andrey Kurenkov, Ashwin Balakrishna, Matthew Matl, David Wang, Roberto Martín-Martín, Animesh Garg, Silvio Savarese, and Ken Goldberg. "Mechanical search: Multi-step retrieval of a target object occluded by clutter." In 2019 International Conference on Robotics and Automation (ICRA), pp. 1614-1621. IEEE, 2019.
> > >
> > > [5] Krainin, Michael, Brian Curless, and Dieter Fox. "Autonomous generation of complete 3D object models using next best view manipulation planning." In 2011 IEEE international conference on robotics and automation, pp. 5031-5037. IEEE, 2011.
> > >
> > > [6] Liu, Yang, Pansiyu Sun, and Akio Namiki. "Target tracking of moving and rotating object by high-speed monocular active vision." IEEE Sensors Journal 20, no. 12 (2020): 6727-6744.

---

### Note · Authors · 2025-08-13

Thank you for the engagement. We were happy to see that all reviewers received our submission with largely positive reviews and provided constructive comments.

We discussed CA6i's concerns on the comparison to search based heuristics and task complexity.  We discussed that our tasks are more complex than prior work in learning-based approaches to active vision. Our manuscript showed that modern generalist VLA policies struggle to solve our tasks, and that our approach can be applied to endow such VLAs with the ability to search, meaningfully improving the state-of-the-art. It also establishes that when heuristic search approaches are possible, our policies substantially outperform them (Tab 2).

Ultimately, hand-coded heuristic active and interactive perception policies are restricted to simple tasks (where they indeed may have advantages since they are unaffected by distribution shifts, as CA6i points out). However, data-driven policies hold the promise of scaling to more complex tasks, and our efforts in this project are an extension of the community’s progress towards data-driven robotic policies, facilitating learning previously challenging (inter-)active perception behaviors. To give proper context to the reader about the space of approaches, we will include these discussion points in our related work and conclusion.

u7RD was interested in the nature of privileged information, and asked about comparisons to potential baselines. 3ct1 also asked about comparisons to a potential baseline, and suggested improvements to presentation. We established in discussions that these baselines are inapplicable in our real-world RL case, due to improper assumptions of the baselines (access to simulation, factorized action space, sample budget, etc.). We also answered their clarification questions, and improved our presentation with their suggestions.

In the background, we have been steadily updating our manuscript based on these reviews and discussions, and it has improved substantially in the process, so thanks again to the AC and reviewers!

---

### Decision · Program_Chairs · 2025-09-17

**Decision:**

Accept (poster)

**Comment:**

This paper presents an approach for RL in partially-observed environments in cases where "privileged" full observations are available during the first phase, and not available afterwards. This problem setting is a natural fit for sim2real settings in which the simulator provides privileged information and the real environment does not. The proposed approach is based on advantage-weighted regression, and adapts it by privileging the advantage function, but not the policy.

Reviewers found that the approach was a meaningful advancement with clear and grounded motivation. Some concerns were raised about the simplicity of the settings not supporting the generalizability of the approach. Concerns were also raised about real-world benefits relative to heuristic methods for adapting to partial observability.

I myself have lingering concerns about the definition of the approach and an important missing baseline. Regarding approach definition: I encourage the authors to clarify in the paper why the privileged information ($o_t^p$) in $o_t^+$ is available in the online phase, as it results in what seems to be a different setting from the motivated setting of the policy not having access to privileged information during online finetuning. Because this information is available throughout both the offline and online phases, it raises the possibility of an approach that simply uses both $(z, o_t^p)$ as input to the policy. This was not tested, instead, the closest method of comparison is the  ablation of $o_t^+$ as input to the critics (denoted "AWR" in S4.2). I therefore think this is an important missing baseline (a policy that also conditioned on $o_t^p$). Without this method of comparison, the performance difference between AAWR and AWR shown in Fig. 3 (and elsewhere) could simply be due to including the extra information $o_t^p$ at all, rather than ensuring that it's only added to the critics. It is possible that adding that information also to the policy results in comparable or superior performance to AAWR, thereby undermining the original formulation of the approach as needing to treat "privileged" information as separate, since the privileged information is in fact available from the environment at all phases of the method.

Relatedly, the theoretical connection between $s$ and $o_t^+$ appears to be missing, and I'll note that $z$ appears nowhere in Alg. 1, yet, it appears as a term in Eq. 3 and the IQL loss for the critics, both of which are part of Alg. 1. I did not see $z$ defined in the implementation details (in either S3.2 or Appendix A).

Given the otherwise positivity of the reviews, my understanding of the paper's contributions, and that the authors can no longer provide feedback to the concerns raised above, I hesitate to recommend rejection on this basis.

---

> ### Public Comment · ~Edward_S._Hu1 · 2025-10-30
> **Clearing up lingering confusions**
>
> We thank the PC for the comments, and address the confusion below.
>
> **Availability of Privileged Observations during Training vs. Deployment**
>
> To be clear, **privileged observations are only available during training time**, and are not available during policy deployment. The training phase includes both __offline and online__ phases. After training, the policy is deployed to the environment where only partial observations are available.  Therefore a privileged policy cannot be run in our deployment setting. To make this clear, we have improved the description of training and deployment in Implementation Details and algorithm blocks in Figure 3.
>
> **Details on $s, o^+$, and policy input $z$**
>
> In practice,  the privileged observation $o^+=(o, o^p)$ serves as an improved estimate of the state:
> $\mathcal{H}(s \mid o^+) << \mathcal{H}(s \mid o)$
> , as it reduces the advantage estimation error. In some cases, we can get perfect (e.g. sim) estimates of the state, or noisier estimates in the real world. Noisier estimates of the state are still useful as privileged inputs, as it reduces the advantage estimation error, and we have verified this through our real world experiments. We leave exact characterization of how privileged information noise affects AAWR to future work. We expect this to be relatively straightforward by leveraging prior theoretical results in asymmetric learning [1].
>
> $z=f(o_1, \ldots o_t)$, the observation history representation for the policy is first defined in Section 3's POMDP description. It is then described in section 3.2, where we introduce the AAWR loss: "We aim to train a policy ... conditioned on the agent state $z_t$ (equivalent to history $h_t=(o_1 \ldots o_t)$".
>
> 1. Lambrechts, Gaspard, Damien Ernst, and Aditya Mahajan. "A Theoretical Justification for Asymmetric Actor-Critic Algorithms." ICML (2025).